# Marine phosphorus and atmospheric oxygen were coupled during the Great Oxidation Event

Matthew S. Dodd [1,2,3,4,13] ✉, Chao Li [2,3,4,13] ✉, Haodong Gu[4], Zihu Zhang[2,3], Mingcai Hou[2], Aleksey Sadekov[1,5], Carlos Alberto Rosière[6], Franco Pirajno[7], Lewis Alcott [8], Frantz Ossa Ossa [8,9,10], Benjamin J. W. Mills [11] & Andrey Bekker[9,12]

The Great Oxidation Event (GOE) represents a major increase in atmospheric $O_2$ concentration between ca. 2430 and 2060 million years ago, culminating in the permanent shift to an oxygenated atmosphere. It's causes remain debated. Here we use the carbonate-associated phosphate (CAP) proxy to reconstruct oceanic phosphorus concentrations during the GOE from globally distributed sedimentary rocks. We find that the CAP and the inorganic carbon isotope composition of marine sediments co-varied during the GOE, suggesting synchronous fluctuations in marine phosphorus, biological productivity, and atmospheric $O_2$. Biogeochemical modelling shows that transient increases in P bioavailability can raise oxygenic primary production and organic carbon burial, yielding isotopically heavy seawater inorganic carbon and reproducing the observed patterns. Consequently, geochemical and modelling data together suggest that P availability was a likely contributor to the rapid oxygenation of Earth during the GOE.

The Great Oxidation Event (GOE) encapsulates the initial permanent rise of atmospheric $O_2$[1]. Accordingly, this was a major event in the evolution of Earth's habitability, which paved the way to Earth's oxygenated atmosphere, the emergence of eukaryotic life and geochemical transformations of Earth's surface and interior[2]. While multiple lines of evidence exist for the GOE[1,3], its beginning has often been defined by the disappearance of mass-independent fractionation of sulphur isotopes (MIF-S) in sedimentary sulphides and sulphates, which represents the transition to $O_2$ concentrations above $10^{-6}$ of the present atmospheric level (PAL)[4]. By this definition, the GOE started at ca. 2430 million years ago (Myr)[5], however, atmospheric $O_2$ is thought to have continued to rise even more rapidly after this initial transition, overshooting to between 1 and 40% of the PAL[6], until an inferred collapse in atmospheric $O_2$ concentrations before 2060 Myr (Fig. 1a-b)[1,3,7–9]. Consequently, the GOE is currently viewed as a protracted event during which $O_2$ level may have fluctuated on a large scale[1,5,10], with potential superimposed shorter and low-level fluctuations, allowing for the possible oscillatory loss and return of MIF-S

[1]School of Earth and Oceans, University of Western Australia, Perth, WA, Australia. [2]State Key Laboratory of Oil and Gas Reservoir Geology and Exploitation & Institute of Sedimentary Geology, Chengdu University of Technology, Chengdu, China. [3]International Center for Sedimentary Geochemistry and Biogeochemistry Research, Chengdu University of Technology, Chengdu, China. [4]State Key Laboratory of Geomicrobiology and Environmental Changes, China University of Geosciences, Wuhan, China. [5]Centre for Microscopy, Characterisation and Analysis, The University of Western Australia, Perth, WA, Australia. [6]Department of Geology, Universidade Federal de Minas Gerais, Belo Horizonte, Minas Gerais, Brazil. [7]School of Earth and Environment, University of Bristol, Bristol, UK. [8]Department of Earth Sciences, Khalifa University of Science and Technology, Abu Dhabi, UAE. [9]Department of Geology, University of Johannesburg, Johannesburg, South Africa. [10]Polar Research Center, Khalifa University of Science and Technology, Abu Dhabi, UAE. [11]School of Earth and Environment, University of Leeds, Leeds, UK. [12]Department of Earth and Planetary Sciences, University of California, Riverside, CA, USA. [13]These authors contributed equally: Matthew S. Dodd, Chao Li. ✉e-mail: matthew.dodd@uwa.edu.au; chaoli@cdut.edu.cn

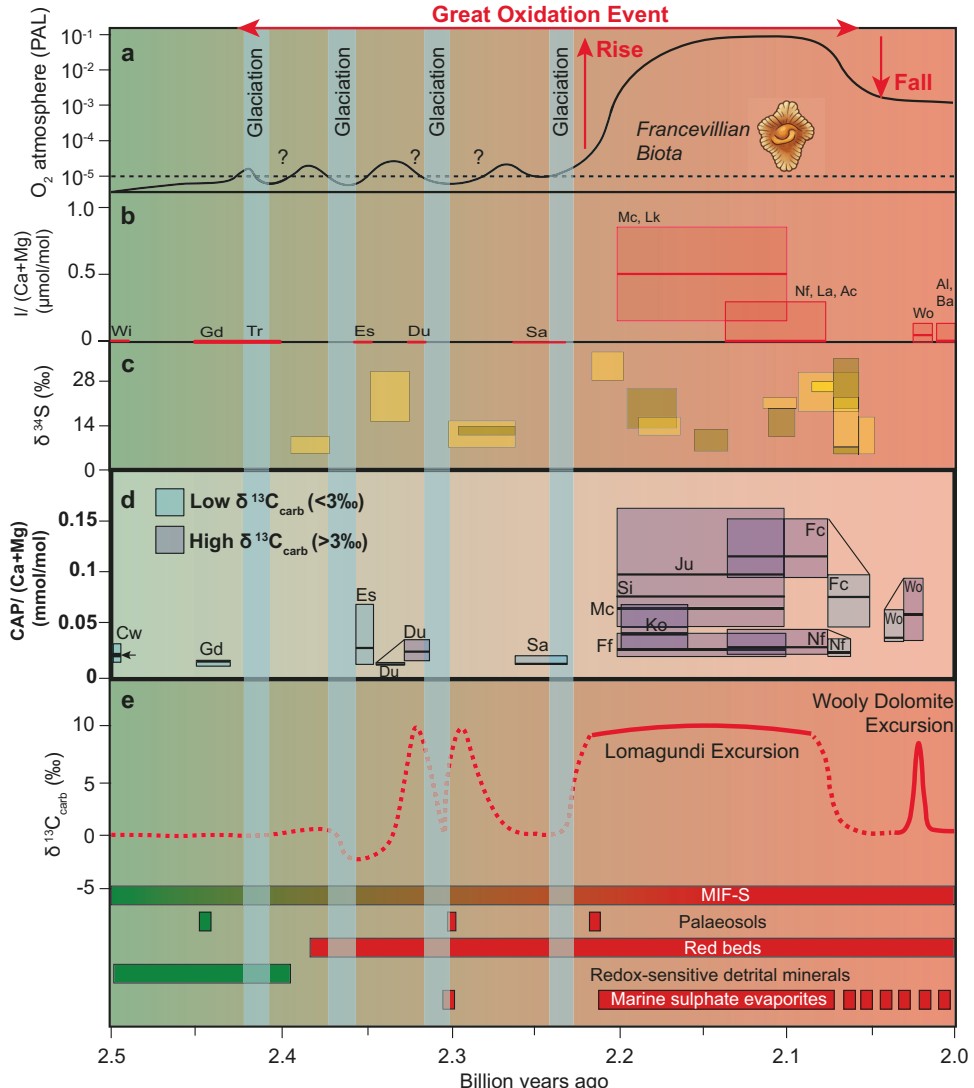

**Fig. 1 | Overview of geochemical changes during the GOE. a** Hypothetical atmospheric oxygen level from this study, fossil images denote approximate first appearance of possible eukaryote and multi-cellular fossils. **b** Box plots (2nd to 3rd quartiles and median line of carbonate-bound iodate values, which is a proxy for dissolved oxygen in seawater from ref. 19. **c** Seawater sulphur isotope composition (yellow boxes from CAS, brown boxes from evaporite) modified after refs. 10, 34. **d** Box plots (2nd to 3rd quartiles and median line) of CAP data from this study, widths depict age uncertainties and tie points between two boxes reflect CAP from a single geological section with high (>3‰; darker shade box) and low (< 3‰; lighter shade box) δ¹³C values. Cw Carawine, Wi Wittenoom, Gd Gandarela, Tr Turee Creek,

Es Espanola, Du Duitschland, Sa Saunders, Ff Fecho do Funil, Si Silverton, Mc Mcheka, Lk Lucknow, Ju Juderina (note Si, Mc and Ju have been grouped into one box plot with one median line (emboldened) for each formation), Fc Francevillian FC formation, La Lower Albanel, Ac Aquas Clara, Nf Nash Fork Formation, Wo Wooly Dolomite Formation, Al Aluminium River, Ba Balser Group. Arrow next to Cw box plot signifies age = ca. 2.6 billion years ago. **e** Idealised seawater inorganic carbon isotope profile and overview of redox proxies (green = reducing and red = oxidising conditions) after ref. 15. Vertical blue bars represent possible glacial intervals.

during the early half of the GOE[5]. These shorter-term fluctuations were likely coupled to the occurrence of global glaciations and changes in seawater sulphate levels between ca. 2430 and 2220 Myr (Fig. 1)[1,7,8,11]. Henceforth in this study, the term 'GOE' refers to the protracted geological interval between ca. 2430 and 2060 Myr.

A unique feature of the later stage of the GOE is the largest and longest-lived positive excursion of sedimentary carbonate-C isotopic composition (δ¹³C$_{carb}$) in Earth history, with an average δ¹³C$_{carb}$ value of +8‰, but highest values reaching up to +28‰[12]. This δ¹³C$_{carb}$ excursion is commonly referred to as the Lomagundi Excursion or Lomagundi Event (LE), which is globally recorded in carbonate sediments deposited between ca. 2220 and 2060 Myr ago (Fig. 1)[13]. The canonical interpretation for the LE is enhanced organic relative to inorganic

carbon burial, and consequently oxygenation of the surface environment. This may have been driven by increased availability of the bio-limiting nutrient, phosphorus (P)[10,13-16]. This interpretation of the LE might be supported by the permanent disappearance of MIF-S at the onset of, or during the LE, which potentially implicates the LE as the ultimate driver to permanent oxygenation of the atmosphere[5]. While it is uncertain whether a net gain in atmospheric O₂ during the LE was responsible for the ultimate disappearance of MIF-S, several lines of evidence suggest that atmospheric O₂ concentrations peaked during the LE and did not reach such levels again until some 1–1.5 billion years later[1,3,17]. Other interpretations for the LE have suggested that increases in primary productivity and organic matter burial were less extreme than inferred by the canonical model, or the δ¹³C$_{carb}$ excursion was not

**Table 1 | Geochemical statistics for elemental and isotope proxies**

| Metamorphic grade | Formation name and (location) | Statistical measure | $\delta^{13}C_{carb}$ vs. CAP | $\delta^{18}O_{carb}$ vs. CAP | Fe vs. CAP | La vs. CAP | Mn/ Sr vs. CAP | Mn vs. CAP | Sr vs. CAP | U vs. CAP | Y/Ho vs. CAP |
|---|---|---|---|---|---|---|---|---|---|---|---|
| Unmetamorphosed | FC (Gab), Juderina (Aus) | $R$ | 0.40 | 0.44 | −0.28 | −0.30 | −0.38 | −0.40 | −0.10 | 0.30 | −0.02 |
| | | $R^2$ | 0.76 | 0.15 | 0.31 | 0.30 | 0.58 | 0.60 | 0.15 | 0.00 | 0.00 |
| | | $t$ | 2.72 | 3.03 | 1.77 | 1.90 | 2.51 | 2.71 | 0.61 | 1.93 | 0.14 |
| | | $p$ | **<0.01** | **<0.01** | >0.01 | >0.01 | >0.01 | >0.01 | >0.01 | >0.01 | >0.01 |
| Prehnite-pumpellyite | Wooly Dolomite (Aus), Carawine Dolomite (Aus) | $R$ | 0.50 | 0.39 | 0.08 | 0.36 | 0.29 | −0.35 | −0.27 | 0.15 | −0.52 |
| | | $R^2$ | 0.25 | 0.15 | 0.01 | 0.13 | 0.08 | 0.12 | 0.07 | 0.02 | 0.27 |
| | | $t$ | 3.93 | 2.88 | 0.56 | 2.65 | 2.08 | 2.54 | 1.92 | 1.04 | 4.12 |
| | | $p$ | **<0.01** | **<0.01** | >0.01 | >0.01 | >0.01 | >0.01 | >0.01 | >0.01 | **<0.01** |
| Greenschist | Mcheka (Zim), Nash Fork (USA), Saunders River (USA), Gordon Lake (Can), Kona Dolomite (USA), Espanola (Can), Fecho do Funil (Bra), Ganderela (Bra), Silverton (SA), Duitschland (SA) | $R$ | 0.23 | 0.17 | 0.07 | 0.14 | −0.23 | −0.27 | −0.20 | 0.03 | −0.15 |
| | | $R^2$ | 0.05 | 0.03 | 0.00 | 0.02 | 0.06 | 0.07 | 0.04 | 0.00 | 0.02 |
| | | $t$ | 3.05 | 2.20 | 0.89 | 1.86 | 3.08 | 3.57 | 2.58 | 0.35 | 1.89 |
| | | $p$ | **<0.01** | >0.01 | >0.01 | >0.01 | **<0.01** | **<0.01** | >0.01 | >0.01 | >0.01 |

All data is derived from CAP partial leaches, except isotopic data, which is obtained from the bulk rock. Statistically significant $p$ values of <0.01 are emboldened.
*Aus* Australia, *Gab* Gabon, *Zim* Zimbabwe, *USA* United States of America, *Can* Canada, *Bra* Brazil, *SA* South Africa.
Bold text is when p values are less than 0.01, signifying statistically significant correlations.

related at all to changes in productivity and organic matter burial, and in turn to P availability (see ref. 18 for a detailed review). Given the number of competing models, whether changes in seawater P availability and associated organic carbon burial are responsible for the rise and fall of atmospheric $O_2$ level during the later part of the GOE remains uncertain. In this study, we use the Carbonate-Associated Phosphate (CAP)[19], a new proxy[19], to reconstruct variability in seawater P levels across the GOE, allowing for the examination of whether changing marine P availability drove the Earth's permanent oxygenation 2 billion years ago.

Carbonate minerals incorporate elements into their crystal structure proportionally to the elemental concentrations in the seawater[20]. Based on this premise, it is possible to estimate ancient marine P concentrations using the concentration of phosphate in carbonate minerals[19,21], when factoring in ambient seawater chemistry (e.g. pH, alkalinity and temperature) and mineralogy, which also play a role in the uptake of elements by carbonate minerals[19,21–23]. In marine sediments, P is predominantly found in phosphate minerals, with additional amounts bound to iron, sorbed to mineral surfaces, or contained within organic matter or carbonate minerals[24]. These P pools, with the exception of carbonates generally, are largely controlled by redox conditions and biological activity[25], and not directly attributable to dissolved seawater P concentration[24]. In contrast, phosphate bound in carbonate minerals (i.e. CAP) can be attributed to seawater dissolved P concentration. In order to extract and measure CAP, a leaching protocol that has been previously developed and validated was used in this study (see 'Methods')[19,22]. This leaching protocol was applied to sediments deposited during the GOE in order to reconstruct relative marine P variations across the GOE. The extracted P was then normalised to Ca and Mg in the leachate to give CAP as mmol of P per mol of Ca and Mg.

We analysed fourteen carbonate formations from four continents (see Table 1 and Supplementary Information 1) to capture whether global seawater phosphate availability changed during the GOE. These carbonate formations capture the LE as well as other shorter-duration positive $\delta^{13}C_{carb}$ excursions during the GOE[26–28]. This provides an opportunity to study coupled changes in $\delta^{13}C_{carb}$ and CAP and, therefore, determine whether P-driven productivity could have triggered the significant rises and falls in atmospheric $O_2$ during the GOE. All of the studied formations were deposited in shallow-water (sabkha

and peritidal, platform) environments, with the exception of the Silverton Formation and the lower and upper Duitschland and Nash Fork formations, respectively, which were deposited in upper-slope and deep-basin settings. From prior work, it was found that dolomite generally hosts larger concentrations of CAP compared to calcite[22], which can bias CAP records via changes in the dominant style of carbonate mineral preserved. However, carbonate mineralogy is almost exclusively dolomite in our sample set, with the exception of calcite in the lower part of the Espanola Formation (see Supplementary Information 1), which limits the degree to which mineralogy affects our CAP trends. The degree of metamorphism is generally uniform across each individual formation, with the exception of the Duitschland Formation, which has experienced local, thermal contact metamorphism in its lower part. Given metamorphism varies amongst formations, we have grouped formations into three metamorphic categories (Table 1), to avoid potential issues regarding CAP preservation at different degrees of metamorphism[29].

## Results

Among all the measured formations, CAP values are on average higher in carbonates with higher $\delta^{13}C_{carb}$ values, when comparing formations from a singular basin, except for the Saunders Formation of the Chocolay Group (ca. 2.3 Ga) and the Espanola Formation (ca. 2.4 Ga) of the Huronian Supergroup on the Superior Craton (Fig. 1c; Table 1 and Supplementary Information Table 1). To fairly compare CAP values, we evaluate studied formations based on their metamorphic grade, which yields a statistically significant ($p < 0.01$) positive Pearson correlation between CAP and $\delta^{13}C_{carb}$, for every metamorphic grade (Fig. 2). This correlation is also supported by continuous chemostratigraphic trends for the FC (ca. 2.1 Ga) and Nash Fork formations (ca. 2.1 Ga) and Wooly Dolomite (ca. 2.0 Ga), which show positive trends in $\delta^{13}C_{carb}$ and CAP for carbonates deposited during the LE and Wooly Dolomite excursion, respectively (Fig. 3). The carbonates from the LE carry the highest CAP values in the dataset reaching values of 0.408 mmol/mol for the Juderina Formation (ca. 2.1 Ga) compared to carbonate deposited prior to the LE with values as low as 0.005 mmol/mol for the Carawine dolomite (ca. 2.6 Ga). On average, CAP is 4, 3.1, 9.2, 2 and 1.6 times higher in carbonates deposited during the LE relative to those that bracket it in South Africa, Brazil, West Australia, Gabon and Wyoming, respectively (Fig. 1c, d). Similarly, carbonates deposited during the

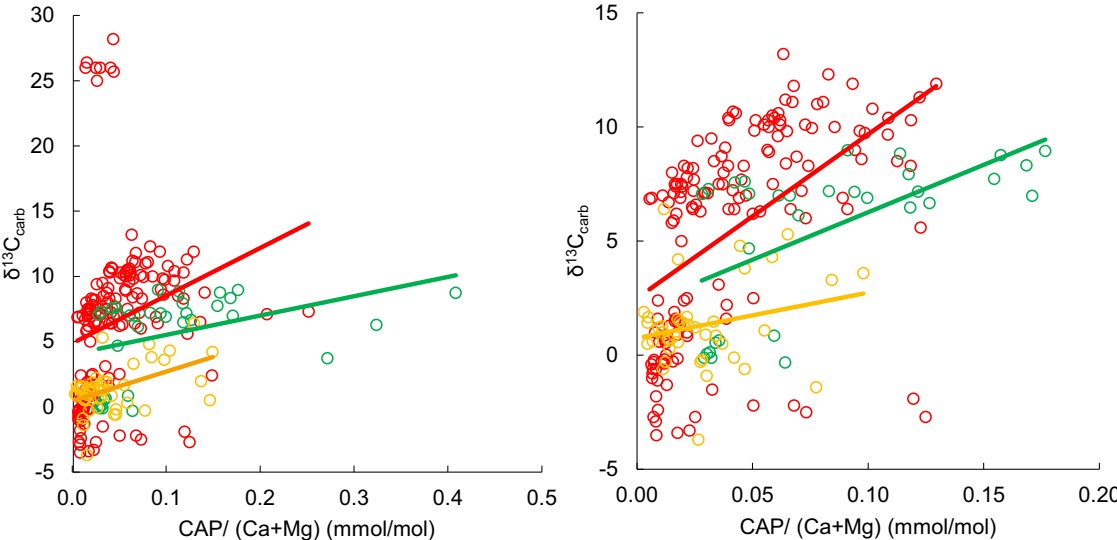

**Fig. 2 | Correlation between CAP/(Ca + Mg) (mmol/mol) and $\delta^{13}C_{carb}$ in the global dataset. a** All data. **b** After removing outliers defined as values beyond 1.5× the interquartile range. Points are coloured by metamorphic facies: green = unmetamorphosed; dark orange = prehnite–pumpellyite red = greenschist. Solid lines are least-squares fits for each facies. **a**: unmetamorphosed $R = +0.4$, $p < 0.01$ prehnite–pumpellyite $R = +0.5$, $p < 0.01$; greenschist $R = +0.23$, $p < 0.01$. **b**: unmetamorphosed $R = +0.61$ $p < 0.01$ prehnite–pumpellyite $R = +0.34$, $p < 0.01$; greenschist $R = +0.51$, $p < 0.01$ Overall, higher $\delta^{13}C_{carb}$ values tend to coincide with higher CAP values.

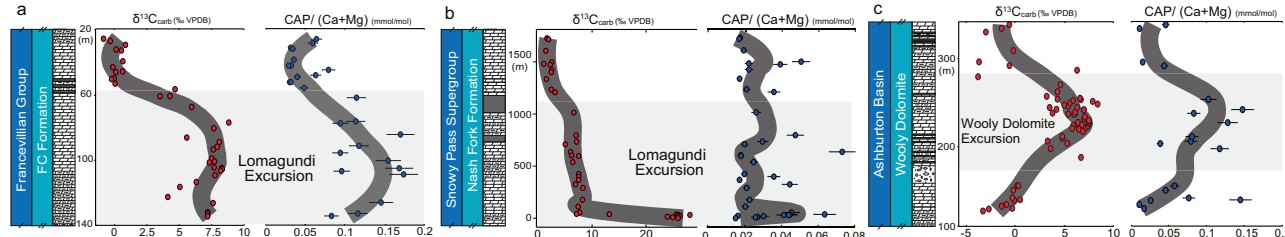

**Fig. 3 | Secular trends in CAP values during the Lomagundi and Wooly Dolomite $\delta^{13}C$ excursions.** $\delta^{13}C$ and CAP data for carbonate successions capturing the Lomagundi excursion at ca. 2100 million years ago from **a** the FC Formation, Gabon and **b** the Nash Fork Formation, Wyoming and **c** the Wooly Dolomite excursion, at ca. 2030 million years ago, from the Wooly Dolomite, Western Australia. Note that the high $\delta^{13}C_{carb}$ values are generally accompanied by the high CAP values. Dark-grey lines represent locally weighted scatter plot smoothing. Error bars reflect the cumulative analytical error uncertainty of 10%. Light-grey boxes define carbonate carbon isotope excursions.

shorter-duration, positive, pre- and post-LE $\delta^{13}C_{carb}$ excursions identified in the Wooly Dolomite, Duitschland and Gordon Lake formations and their equivalents have average CAP values that are 2, 2.3 and 4.5 times higher in samples, respectively, compared to carbonates with near-to-zero C isotope values within these units or stratigraphically close to them (Figs. 1 and 3). For example, for the Wooly Dolomite, CAP reaches 0.13 mmol/mol during the peak of the $\delta^{13}C_{carb}$ excursion and drops as low as 0.011 mmol/mol in carbonates not recording the $\delta^{13}C_{carb}$ excursion.

## CAP data evaluation and discussion

CAP can be used to directly estimate relative oceanic phosphate level at the time of carbonate precipitation, if detrital contamination, diagenetic/metamorphic alteration and other potential modifications to the primary seawater signature can be screened out[19,29]. Our CAP extractions show no sign of contamination with other sources of sedimentary P, as there is little to no correlation between CAP and La (an element used for tracking phosphate mineral dissolution[19]; Table 1). It has previously been proposed that the LE and other, positive short-lived $\delta^{13}C_{carb}$ excursions during the GOE may represent diagenetic alteration[30]. In order to reach the positive $\delta^{13}C_{carb}$ values of the LE, it was suggested that methanogenesis in the sedimentary pile contributed with residual isotopically heavy carbon species, which

were then incorporated into carbonate minerals[30]. If true, these carbonates formed in the methanogenic zone would be enriched in CAP due to remineralisation of organic matter, similar to the Miocene Monterey Formation concretions[19] that have a large range of $\delta^{13}C_{carb}$ values consistent with intense methane production and high CAP values. However, there are several lines of evidence against an overarching diagenetic control, including methanogenesis: (1) An important geochemical feature of carbonates formed in the methanogenic zone would be low concentrations of Carbonate-Associated Sulphate (CAS) with highly positive $\delta^{34}S_{CAS}$ values due to the Rayleigh distillation with near-complete reduction of all available sulphate in the methanogenic zone[31]. However, CAS concentrations are high in carbonates deposited during the LE[8,32] and $\delta^{34}S_{CAS}$ values are inversely correlated with $\delta^{13}C_{carb}$ values in the LE Mcheka Formation, Zimbabwe, arguing against carbonate formation in the methanogenic zone[8]. Moreover, high CAS concentrations during the LE are consistent with a relatively larger seawater sulphate reservoir at that time, which is supported by S isotope data recorded in carbonate and coeval black shales deposited on multiple cratons[33]. (2) Importantly, the $\delta^{34}S_{CAS}$ values in the LE carbonates and coeval sulphate evaporite deposits are roughly equivalent (Fig. 1b), suggesting that the analysed carbonates generally capture the chemical signal of contemporaneous seawater, supporting the premise that CAP signals may also be faithful records of

seawater P[8,34] (see Supplementary Information text 2). (3) The LE Mcheka Formation carbonates (Supplementary Information 1) have I/(Ca+Mg) values above 0.5 µmol/mol[17], suggesting precipitation from oxygenated waters, which contrasts with authigenic carbonates of the Miocene Monterey Formation, which formed in the methanogenic zone of the anoxic sedimentary pile[35]. (4) There is no significant correlation between Mg/Ca and Mn/Sr, which are commonly used as tracers of mineralogical controls and late diagenetic meteoric effects on carbonate chemistry, respectively[36]. (5) Samples from the Francevillian FC Formation, Gabon, have been previously extensively screened for diagenetic alteration using trace elements and petrography[37], which also supports a primary origin for the CAP and $\delta^{13}C_{carb}$ trends (Fig. 3a). (6) There is no statistically significant correlation between CAP and U or Fe concentrations in the carbonate leaches (Table 1), suggesting that diagenetic alteration by changing redox conditions in pore water or post-depositional fluids is not likely to have been responsible for CAP values in the analysed samples. We do acknowledge, however, the presence of significant point-to-point scatter in the CAP data within individual units and among formations (Fig. 3), which is common in Precambrian carbonate records due to a combination of lithological heterogeneity, analytical noise and potential post-depositional alteration. However, the replicated trends across the Francevillian and Snowy Pass basins and the stratigraphic coherence among globally correlative carbonate successions suggest that the broader patterns reflect primary seawater signals.

Aside from diagenesis, shifts to positive $\delta^{13}C_{carb}$ values could result from the diurnal carbon cycle, as observed for modern-day Bahamian carbonates[38], or other local processes such as seawater evaporation. However, we note that a diurnal carbon cycle is unlikely to produce coupling between CAP and $\delta^{13}C_{carb}$ values, but rather the opposite. This is because a diurnal cycle results in high $\delta^{13}C_{carb}$ values during periods of enhanced photosynthesis, which would draw down P. In addition, evaporitic conditions could induce noticeable shifts to $\delta^{13}C_{carb}$ and CAP in shallow-marine settings, due to restriction from the open ocean, whereby implying that the observed $\delta^{13}C_{carb}$ and CAP trends reflect local effects from globally distributed, isolated regions from the global ocean basins[39]. While a plausible explanation, there are a number of conflicting observations, such as the occurrence of high and/or positively correlated trends in $\delta^{13}C_{carb}$ and CAP in inferred open, deep-marine settings such as the Nash Fork and Silverton carbonate successions deposited during the LE. However, in the case of the deep-marine Silverton Formation, some have speculated that these sediments might reflect turbidite deposits of shallow-marine sediments[40]. Moreover, there is no statistically significant correlation between CAP and Y/Ho (a commonly used indicator of basinal restriction and water mass origins), with the exception of the Wooly Dolomite and Carawine Dolomite (Table 1). If evaporitic conditions did diagenetically alter CAP or detach local marine P levels from the global ocean P reservoir, we could expect to see some correlation between CAP and tracers of different water masses.

Alternatively, there were likely significant changes in ocean chemistry over the 600 Myr during which the samples analysed in this study were deposited[41], which could have changed P incorporation into carbonate without a change in seawater P concentration[19,22]. A suite of controlled precipitation experiments found that CAP displays a negative relationship with pH, alkalinity and carbonate precipitation rate and a positive relationship with temperature. To explain the increase in CAP values during the LE and other carbon isotope excursions during the GOE as a result of changes in seawater chemistry alone, seawater chemistry would need to shift to lower pH and alkalinity, which could also decrease carbonate precipitation rate. The combined effects of lower pH, alkalinity and precipitation rate could result in increased P incorporation into carbonate. Alternatively, seawater temperature would need to increase to force high P incorporation into carbonate. Temperature exerts a relatively small effect on

CAP, with a 20 °C change resulting in an increase in CAP by a factor of 1.5[19]. While a decrease in pH from 8 to 7 could increase CAP by a factor of 2, and a decrease in alkalinity by a factor of 4 could increase CAP by a factor of 1.7, a decrease in precipitation rate of 95 times would result in a CAP increase by only a factor of 1.25. Given that half of our sample sets exhibit CAP shifts by more than a factor of 4 across the positive $\delta^{13}C_{carb}$ excursions, an exceptionally large change in seawater chemistry is required such as a combined large pH change (>1 unit), a 4× reduction in alkalinity and a 95× decrease in carbonate precipitation rate to achieve a more than 4× change in CAP values. Based on current models for marine pH over Earth history, such a pH change is unlikely to occur over the GOE interval[41]. Consequently, it can be reasonably concluded that changing background ocean chemistry (i.e. species other than dissolved phosphate) was not an overarching control on P incorporation into carbonate minerals for these $\delta^{13}C_{carb}$ excursions.

There is a large spread in CAP values in formations deposited during the LE (Fig. 1c), which could be expected given the heterogenous nature of P concentration in the modern surface ocean[42]. An additional factor that might be responsible for the variability in CAP values is the different metamorphic grade of the formations studied, given that metamorphic alteration could lead to CAP loss[29]. This is supported by LE carbonates from the unmetamorphosed and exceptionally well-preserved Francevillian FC and Juderina formations having 1.9 and 2.1 times higher average CAP values compared to the rest of the LE carbonate formations, which have undergone metamorphism at the greenschist facies (Fig. 2). However, the Carawine Dolomite has some of the lowest CAP and $\delta^{13}C_{carb}$ values from our dataset and is of a significantly lower metamorphic grade than most formations within the dataset (Table 1), suggesting that CAP loss during metamorphism is not likely to be a primary control on the coupled CAP and $\delta^{13}C_{carb}$ trends. Additionally, the correlation coefficient of CAP and $\delta^{13}C_{carb}$ values declines with increasing metamorphic grade (Table 1 and Fig. 2), suggesting that metamorphic alteration is an unlikely cause for the positive correlation of CAP and $\delta^{13}C_{carb}$.

Despite greatly varying preservational conditions (neomorphism to greenschist facies and contact metamorphism; see Supplementary Information 1) and in turn likely alteration of primary CAP signals, the general first-order trend to higher CAP in carbonates with higher $\delta^{13}C_{carb}$ values is consistent across all globally distributed basins (Table 1; Fig. 1c; Figs. 2, 3). The array of depositional environments across which these trends are found (e.g. sabkha and peritidal, platform to upper slope and deep basin; see Supplementary Information 1), in conjunction with the global distribution of the studied successions, and the observation of multiple $\delta^{13}C_{carb}$ and CAP excursions over a 600 Myr period during the GOE would require a high degree of coincidence for local or diagenetic processes to produce first-order trends in $\delta^{13}C_{carb}$ and CAP values across such an array of spatiotemporal settings. Nevertheless, the possibility that preservational and sampling biases could have resulted in the analysed successions capturing local or diagenetic effects that culminated in coupled $\delta^{13}C_{carb}$ and CAP values cannot be conclusively eliminated, but based on current observations, this also requires a multitude of assumptions to explain the existence of correlated $\delta^{13}C_{carb}$ and CAP values across such an array of metamorphic grades, carbonate minerals (e.g. calcite and dolomite both preserve elevated $\delta^{13}C_{carb}$ and CAP values in the Silverton Formation), environments of deposition and time periods. On that basis, the most parsimonious interpretation suggests that dissolved phosphate concentrations were elevated in global marginal-marine basins and oceans during the LE and other, short-lived positive $\delta^{13}C_{carb}$ excursions during the GOE (see Fig. 1 and Supplementary Information 2).

## Phosphorus cycling during the Great Oxidation Event

The elevated CAP values observed during the LE and other positive $\delta^{13}C_{carb}$ excursions across the GOE provide strong evidence for

transient growth in the marine P reservoir at that time. These data support a model in which enhanced P bioavailability stimulated high rates of primary productivity, ultimately driving the rapid accumulation of atmospheric $O_2$ at least during the LE. However, interpreting these P trends requires acknowledging that the Precambrian P cycle operated under fundamentally different oceanographic and redox conditions than those in the modern oceans.

Recent work has shown that P and ocean anoxia were sometimes decoupled in Precambrian systems[22], in contrast to their tightly coupled behaviour in modern oxygenated oceans. This decoupling suggests that the feedbacks linking redox conditions, P recycling and productivity may have been weaker or operated differently in deep time. Supporting this interpretation, elevated P burial efficiency has been documented under ferruginous anoxic conditions during the Proterozoic, with both ferruginous and euxinic settings showing higher P to organic carbon burial ratios than those observed in modern sediments[43,44]. These findings indicate that anoxic conditions did not necessarily promote P regeneration and may have contributed to sustained P burial, weakening the regulatory link between marine productivity and redox conditions.

Unlike the modern P cycle, which is strongly coupled to oxygen-rich and sulphate-abundant environments that support efficient recycling of organic-bound P[24], the Precambrian ocean was dominated by ferruginous and sulphate-poor conditions[45–47]. Under these conditions, both P scavenging by reactive iron minerals and inefficient remineralization due to limited sulphate availability would have suppressed marine phosphate concentrations. Our CAP evidence from the Lomagundi interval, however, points to a period when these constraints were relaxed, allowing for a significant expansion of the marine P reservoir. This relaxation likely resulted from the expansion of marine sulphate availability, potentially driven by oxidative weathering associated with tectonic uplift during the Turee Creek Orogeny[13,48,49]. Rising sulphate levels would have stimulated microbial sulphate reduction and enhanced P recycling from organic-rich sediments[8]. P speciation data from this interval support increased recycling efficiency as a result of increasing sulphate availability, due to sedimentary sulphur cycling[50], reinforcing a positive feedback in which rising atmospheric oxygen levels increased sulphate availability in anoxic marine seawater, amplifying P release and sustained high productivity. This feedback likely reached a tipping point during the LE, with high CAP values reflecting peak P bioavailability and export production. However, the subsequent decline in atmospheric oxygen levels suggests that this system was inherently unstable. As oxygen levels rose, Earth system feedbacks may have shifted toward modern-like P cycling, in which oxygenated bottom waters enhance P trapping in sediments through iron and calcium phosphate formation. This shift is supported by the widespread deposition of phosphorites near the end of the GOE, around 2 billion years ago, which typically require suboxic to oxic conditions to form[24]. The temporal coincidence of phosphorite formation with peak CAP values and rising oxygen suggests that enhanced sedimentary P drawdown may have emerged as a stabilising feedback, reducing marine phosphate concentrations and triggering the collapse of atmospheric oxygen levels following the LE. Future work integrating CAP with Total Organic Carbon, total P and trace-metal datasets will help further disentangle P cycling controls on CAP and marine P availability.

Together, these observations indicate that the Precambrian P cycle experienced fundamental transitions in response to evolving redox conditions and that brief windows of enhanced P bioavailability could drive major atmospheric shifts in $O_2$ levels. The CAP record, when interpreted in light of these redox and sedimentary processes, captures the dynamic and sometimes unstable coupling between marine nutrients and Earth's oxygenation trajectory. This interpretation is also consistent with triple-oxygen isotope values of sulphate, which indicate a major increase in primary productivity and $O_2$

production across the GOE and LE[51]. While other models have been proposed to explain positive $\delta^{13}C_{carb}$ excursions during the GOE[16,30,52,53], most models rely on higher seawater P concentrations during positive $\delta^{13}C_{carb}$ excursions across the GOE as a contributor to elevated biological productivity and atmospheric $O_2$ level, which is for the first time demonstrated by the CAP data presented herein.

Despite there being a positive correlation of CAP and $\delta^{13}C_{carb}$ during the GOE, it does not imply that elevated marine P level was the sole driving force behind the GOE, and other forcings could have included a change in the flux of reductants to Earth's surface, e.g. changes in volcanism[54], methane[55] or $H_2$ production[56]/$H_2$ escape to space[57], a switch between two bi-stable redox states[58], or a switch from ecological dominance of anoxygenic to oxygenic photosynthesis[59,60]. These mechanisms can be broadly grouped into a reductant or P-driven GOE, a reductant-driven GOE is one in which $O_2$ sinks gradually diminish and $O_2$ levels gradually rise until reaching some tipping point, allowing for a rapid rise in $O_2$ levels. A P-driven GOE is one where additional P input provides a temporary increase in $O_2$ production. Both could have resulted in a rapid rise in $O_2$ level once $O_2$ production overcame $O_2$ sinks. In order to explore which hypothesis is more likely, we will use a biogeochemical model to assess both potential styles for the GOE.

## Biogeochemical modelling

We utilise a recent global biogeochemical model[61], which has been previously used to quantify the dynamics of P, C and O in the oceans and atmosphere over the course of Earth's history, and has been validated against a suite of long-term geochemical data. We used the model to quantify the dynamics of P, C and O during a hypothetical P-driven oxygenation event that lasted for 200 Myr, which is equivalent to the potential duration of the LE (see 'model description' in 'Methods'). In this model, the global ocean is divided into four regions, representing the proximal and distal continental shelves, and the surface and deep oceans, which are connected and mixed under modern-ocean water-mass fluxes[61] (Fig. 4). We begin the model in a pre-GOE state (as in ref. 43) and modify it to simulate an increase in P input to the oceans from continental weathering during the LE. We used a P flux of sufficient magnitude to reproduce an inorganic carbon isotope excursion of +8‰, which is the average value for the LE (Fig. 1d); the P flux used is yet one to two orders of magnitude lower than modern-day riverine P flux[24]. We note that other complimentary models, should as enhanced carbonate subduction and degassing could generate a positive excursion in $\delta^{13}C$ of marine dissolved inorganic carbon (DIC) ($\delta^{13}C_{DIC}$), in addition to P-driven organic carbon burial, and therefore lower P fluxes may achieve the same result when factoring in these other mechanisms see refs. 16,52. We ran the model 1000 times using a range of input variables (see 'Methods') in a Monte Carlo setup and report the results within a 95% confidence window. Within this window, additional P input increases primary productivity and therefore $O_2$ production, resulting in a geologically rapid rise in atmospheric $O_2$, reaching between 40 to 110% PAL (Fig. 5a, b). Elevated primary productivity results in excess organic carbon burial, driving an increase in global seawater $\delta^{13}C_{DIC}$ values (Fig. 5c–f). High organic carbon burial at this time is consistent with abundant occurrences of deep-water black shales during the LE[62].

In our modified model, we estimated $\delta^{13}C_{DIC}$ values in each ocean region by modifying the global value in line with local relative dissolved P concentrations (see 'Method' description). This 'Redfield slope' method applies an inverse relationship between $\delta^{13}C_{DIC}$ and P concentration based on the modern ocean[63]. Our model results show that in the shallowest regions of the ocean (proximal regions) $\delta^{13}C_{DIC}$ values increase to between +3 and +8‰ due to high rates of photosynthesis, relative to respiration, which results in low dissolved P concentrations (Fig. 5d). The higher end of these $\delta^{13}C_{DIC}$ values is consistent with average $\delta^{13}C_{carb}$ value of the LE carbonates. In the deep

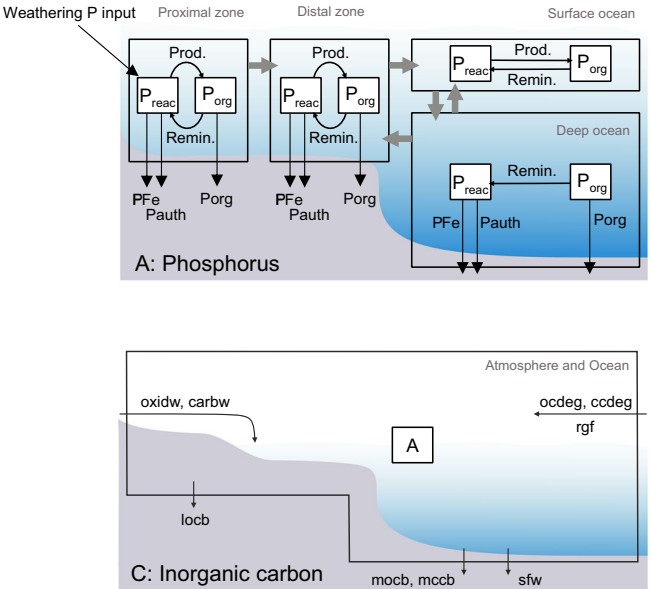

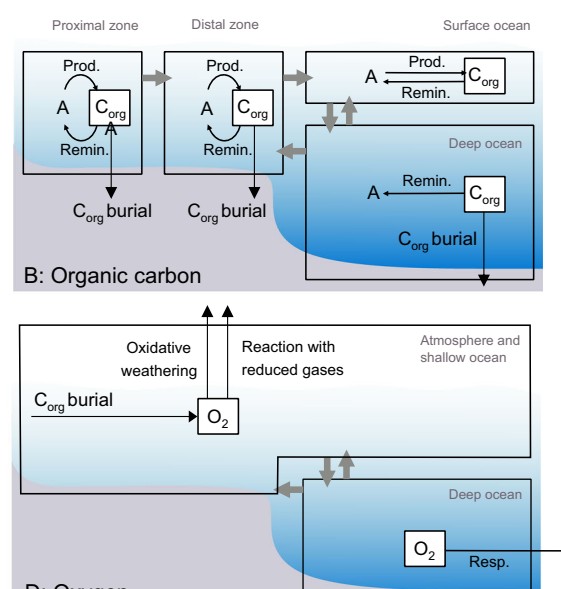

**Fig. 4 | Illustration of the four-box biogeochemical model used in this work.** A Model phosphorus cycle divided into four ocean regions, proximal, distal, surface and deep ocean. $P_{reac}$ dissolved inorganic phosphorus, $P_{org}$ organic-bound phosphorus (i.e. biomass), PFe iron-bound phosphorus burial, $P_{auth}$ authigenic P burial (e.g. apatite), $P_{org}$ organic-bound phosphorus burial. **B** Model of organic carbon cycle, which is also divided into the same 4 ocean regions. A dissolved inorganic carbon, $C_{org}$ organic carbon (i.e. biomass) and $C_{org}$ burial organic carbon burial. **C** One box inorganic carbon cycle for the ocean and atmosphere; oxidw, carbw and sfw oxidative, carbonate and seafloor weathering, respectively. Locb and mocb land and marine organic carbon burial, respectively. Mccb marine carbonate carbon burial, ocdeg organic carbon degassing, ccdeg carbonate carbon degassing and rgf reducing gas flux. Prod. productivity and Remin. remineralisation. **D** Model oxygen cycle. Grey arrows in all figures reflect water mass exchange between boxes and transfer of its chemical contents. See 'methods' and refs. 62,67. for full description.

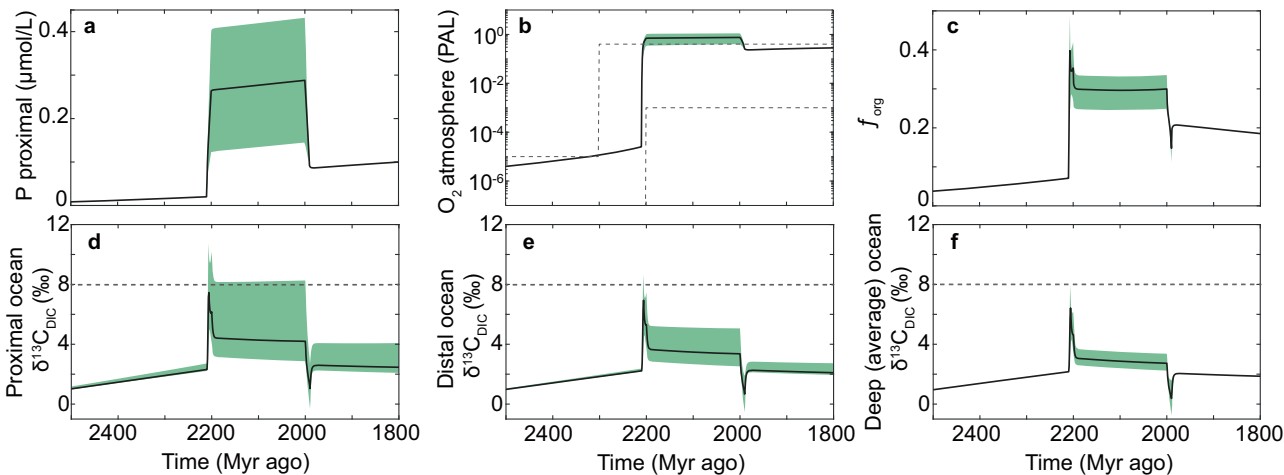

**Fig. 5 | Four-box biogeochemical model results. a** P concentration in the proximal ocean box. **b** Atmospheric $O_2$ concentration. **c** Fraction of inorganic carbon that is buried as organic carbon ($f_{org}$). **d**, **e** Isotopic composition of DIC ($\delta^{13}C_{DIC}$) in the proximal, distal and deep- ocean boxes. PAL present atmospheric level. DIC dissolved inorganic carbon. Green shading reflects model sensitivity range for 95% of results. Dashed grey lines in panel b indicate geochemical proxy evidence for atmospheric $O_2$ level and average $\delta^{13}C_{carb}$ value for the Lomagundi Event.

ocean, $\delta^{13}C_{DIC}$ values only reach around +3‰ due to higher rates of respiration relative to photosynthesis, which favour the build-up of $^{13}C$-depleted, DIC and P (Fig. 5e). In short, our model predicts a range from small (<0.5‰) to large (>5‰) transient $\delta^{13}C_{DIC}$ gradients in the oceans during the LE, as a result of elevated marine P concentration and enhanced primary productivity. This model result could explain observations of the facies-dependent expression of the LE, where the highest $\delta^{13}C_{carb}$ values occur in shallow marine carbonate and smaller $\delta^{13}C_{carb}$ values in purported deeper marine carbonate deposited during the LE[39]. The existence of transiently large, marine $\delta^{13}C_{DIC}$ gradients during the LE could also lower the required amount of biomass burial

needed to explain the LE and prevent extremely high atmospheric and deep-ocean $O_2$ levels (3.0 and 2.5 times modern levels, respectively) predicted by the model, if the whole ocean would have shifted in $\delta^{13}C_{DIC}$ values to +8‰ (i.e. without $\delta^{13}C_{DIC}$ gradient in the oceans, see Fig. 6). However, we note that there are few constraints on Paleoproterozoic P weathering input, marine DIC concentration and the relative content of C and P in primary producers (see 'Methods'), which together prevent constraining marine $\delta^{13}C_{DIC}$ gradients during the LE, until these parameters are better quantified.

After this additional P input to the ocean is removed, organic productivity and $O_2$ production decline, resulting in a drop in

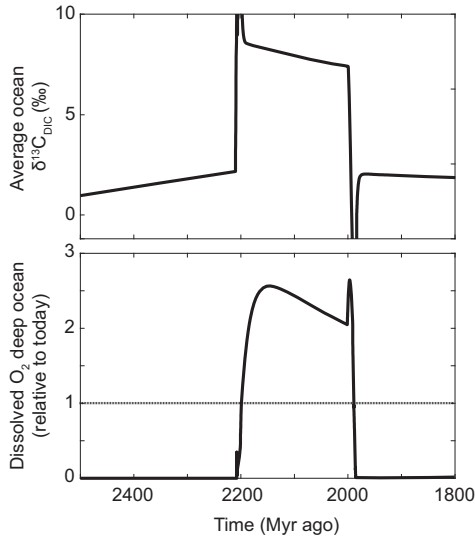
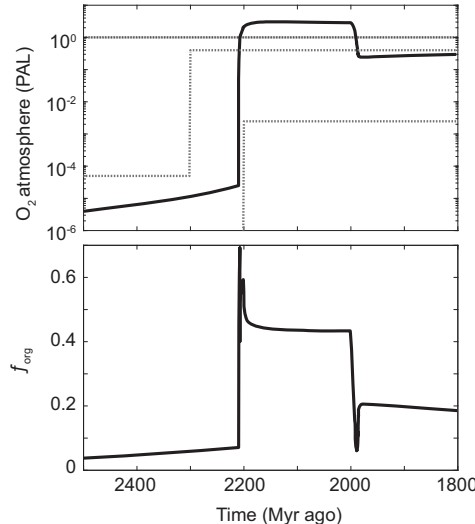

**Fig. 6 | Biogeochemical model results with no dissolved inorganic $\delta^{13}C_{DIC}$ gradient (one box inorganic carbon cycle). a** Average $\delta^{13}C$ of DIC in the ocean. **b** Atmospheric $O_2$ concentration, **c** Dissolved $O_2$ in the deep ocean relative to present oceans, **d** Fraction of inorganic carbon that is buried as organic carbon. PAL present atmospheric level. Grey dashed lines are redox proxy limits for atmospheric $O_2$ level, see ref. [62].

atmospheric $O_2$ level to around 20% PAL (Fig. 5b). These declines in biological productivity and atmospheric $O_2$ level are consistent with geochemical proxies invoking a collapse in primary productivity and atmospheric $O_2$ level after the LE[9,51,64,65]. Note, the absolute atmospheric $O_2$ level in the model is dependent on the riverine P flux, mantle degassing and weathering rates, which are poorly constrained for the Paleoproterozoic and therefore model estimates for atmospheric $O_2$ can vary significantly within its uncertainty range[61]. Nevertheless, model results show that an increase in marine P concentration can feasibly result in a major rise in seawater $\delta^{13}C_{DIC}$ values and atmospheric $O_2$ level, consistent with available $\delta^{13}C_{carb}$, CAP and atmospheric and ocean redox proxy data for the GOE (Fig. 1b). What is particularly noteworthy is when the model is run without an additional P input during the GOE but with a declining reductant flux over time, a surface oxygenation event does occur when oxygen production overwhelms the reductant flux[61,66], however, there are distinct differences between a P-driven GOE and a reductant-driven GOE (Extended Data Fig. 7). Firstly, no appreciable $\delta^{13}C_{DIC}$ excursion is generated in the reductant-driven GOE, whereas a positive $\delta^{13}C_{carb}$ excursion is seen in a P-driven GOE. Second, during a reductant-driven GOE, atmospheric oxygen rises to a new stable level without any subsequent decline in atmospheric $O_2$, whereas during the P-driven GOE, $O_2$ level declines once the change in P input is removed. This decline in atmospheric $O_2$ level after the GOE is consistent with many geochemical proxies[8,9,13,64,67], recording lower seawater oxidant and $O_2$ atmospheric availability immediately after the GOE (Fig. 1b).

Current geochronological age constraints prevent conclusive resolution of whether the LE and other, shorter-lived positive $\delta^{13}C_{carb}$ excursions during the GOE were globally synchronous events. Given the abundance of globally distributed basins that carry large positive $\delta^{13}C_{carb}$ excursion with an age between ca. 2.22 and 2.06 Ga, the likelihood of selectively preserving shallow-marine basins enriched in $\delta^{13}C_{DIC}$ and P due to a series of unrelated events during a 160 Myr window of the LE is low, especially considering that the preservational potential of sedimentary rocks declines with their ages. Therefore, this time period was likely characterised either by numerous, highly productive sedimentary basins or one or multiple events with globally elevated seawater $\delta^{13}C_{DIC}$ values and P concentrations; both scenarios would result in elevated global productivity and $O_2$ production. We prefer the latter scenario because carbonates from inferred, open-marine settings preserve $\delta^{13}C_{carb}$ excursions well exceeding 2–3‰[39,68].

This is only feasible in our model (Fig. 5f) with a global-scale increase in organic carbon burial (i.e. $10^{13}$ mol C/yr), which results in atmospheric $O_2$ level consistent with redox proxy records (i.e. a transition from $<10^{-5}$ to $>10^{-3}$ PAL) (Fig. 5b). Based on our model, this means that the LE and associated rise in atmospheric $O_2$, required a global-scale increase in P availability and in turn in productivity to explain the $\delta^{13}C_{carb}$ record.

## Summary

In conclusion, the positive correlation of $\delta^{13}C_{carb}$ and CAP across multiple time intervals and depositional environments throughout the GOE provides strong evidence for a contemporaneous increase in seawater P level and, in turn, primary productivity and organic carbon burial between ca. 2.43 and 2.06 Gyr ago. However, uncertainties remain regarding the spatial extent to which this positive correlation may have applied (e.g. shallow-marine basins versus global oceans), and how much of the relative change in CAP values reflects changes in seawater dissolved P concentration, as opposed to compounding effects from broader shifts in seawater chemistry. Nevertheless, biogeochemical model results, in addition to proxies for globally elevated primary productivity (e.g. triple-oxygen isotope composition[51]) and evidence for the rapid rise and fall of atmosphere-ocean oxygenation (e.g. redox-sensitive elements and their isotope compositions[5,8,9,17,65,67]) during the LE and other intervals throughout the GOE, converge on a unifying hypothesis. Together, they strongly support the interpretation that P bioavailability was the most likely driver of rapid and large-magnitude fluctuations in atmospheric $O_2$ levels during the later part of the GOE, ending in the permanent oxygenation of Earth's surface, paving the way for the evolution of oxygen-dependent, eukaryotic life at that time, such as the first appearance of *Grypania Spiralis* and the *Francevillian biota* during the LE (Fig. 1a) and afterwards[69]. These events were likely preceded by a long-term, gradual rise in atmospheric $O_2$ level, which may have finally triggered the observed spike in marine P availability and large-scale phosphorite deposits during the later stages of the GOE[49].

## Methods

### Total inorganic carbon (TIC) analysis

TIC was measured using a FOGL Digital Soil Calcimeter. TIC was measured directly by weighing out ~0.2 g of rock powder into gas-tight 250 mL glass bottles and 4 mL of 6 M HCl in a plastic cuvette was

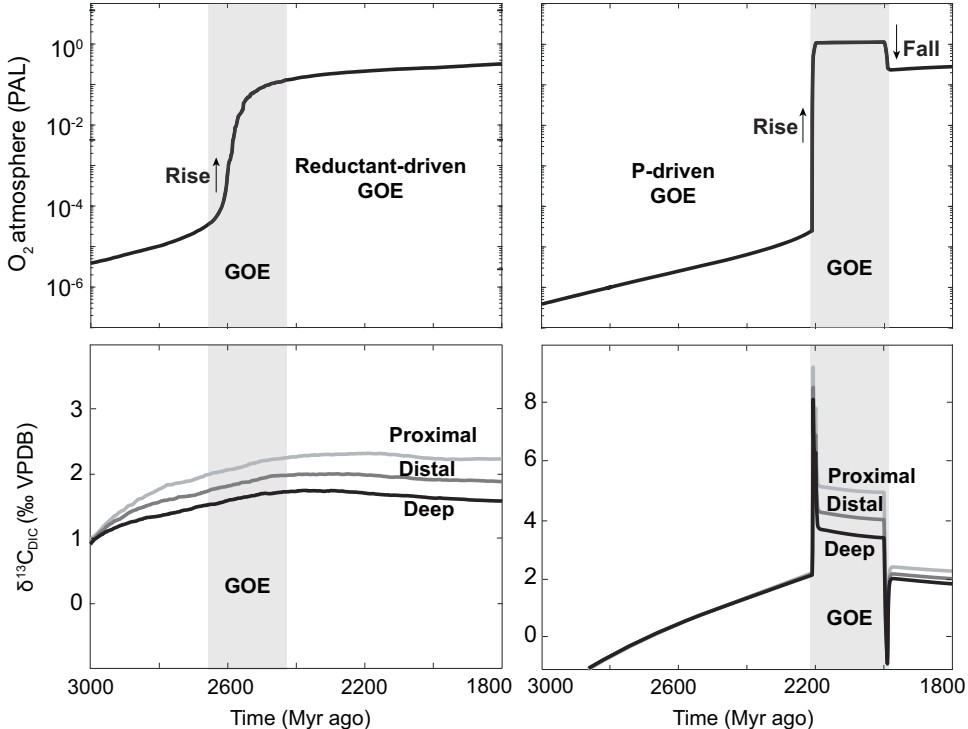

**Fig. 7 | Biogeochemical model results comparing a reductant and P-driven GOE.** **a** Reductant-driven GOE and **b** P-driven GOE showing atmospheric O₂ profiles over time. GOE interval demarked by grey shading. Note, in the P-driven GOE a fall in atmospheric O₂ occurs after the initial rise in atmospheric O₂. **c** carbon isotope value of dissolved inorganic carbon ($\delta^{13}C_{DIC}$) in different modelled ocean regions (proximal, distal, deep) in a reductant-driven GOE and d) P-driven GOE. Grey shading reflects the interval of atmospheric O₂ rise termed GOE.

carefully added to the bottle. Once a bottle was sealed, it was tipped to mix carbonate and HCl and periodically shaken until a stable reading was achieved on the calcimeter. A pure calcium carbonate carbon standard was analysed after every 10 samples to check reproducibility (±2% $CaCO_3$; $n = 30$).

## Carbonate carbon- and oxygen-isotope analysis
Carbonate carbon- and oxygen-isotope analysis was performed in the West Australian Biogeochemistry Centre at the University of Western Australia and Nanjing University. About 60–300 μg of sample powder was loaded into a vial after drying the powders at 70 °C for 24 h in an argon atmosphere. The samples were then reacted with 100% phosphoric acid under a vacuum at 70 °C using a GasBench II or a Kiel IV device. The resulting $CO_2$ was subsequently introduced into a Thermo-Fisher Scientific Delta XL mass spectrometer at the University of Western Australia or MAT 253 mass spectrometer at Nanjing University for isotopic measurements. Delta values were calibrated relative to IAEA reference standards NBS19 and NBS18, and Chinese national standard GBW04416. Carbon-isotope and oxygen-isotope data for carbonates are reported relative to Vienna Pee Dee Belemnite with a precision better than ±0.1‰ and ±0.15‰ (1σ) for carbon and oxygen isotope values, respectively, based on duplicate analyses of standards and unknowns.

## Carbonate-associated phosphate (CAP) analysis
CAP was measured following previously established protocols[19]. Only samples comprising >50% carbonate were chosen for analysis. Based on sample TIC and carbonate mineralogy, sufficient sample powder was weighed out, corresponding to -1 mmol of carbonate. Samples were then repeatedly washed for 24 h, each time using 4 mL of 10% NaCl solution buffered to pH 8 with $NaHCO_3$ to remove adsorbed P. Once adsorbed P was below 0.1 ppm in the wash solution, an appropriate amount of 2% vol/vol acetic acid was added to the powder to dissolve up to 70% of the carbonate. After 30 min the sample was centrifuged, and the leachate was extracted using 0.2-micron polyethersulfone membrane filters. An aliquot of leachate was taken for major and trace element analysis and measured using a Thermo Fisher Element XR Inductively Coupled Plasma Mass Spectrometer at the Centre for Microscopy, Characterisation and Analysis at the University of Western Australia. Analytical errors were better than ±2% for all elements based on duplicate analyses of two internal standards. Another aliquot of the leachate was taken, and the P concentration was determined spectrophotometrically using the malachite green method at 663 nm and a Perkin-Elmer EnSight® plate reader at the University of Western Australia and Chengdu University of Technology, with a relative standard deviation of less than ±5%. The sample residue was then washed with 4 mL of 10% NaCl solution buffered to pH 8 with $NaHCO_3$ for 24 h, and P in the wash was measured using the malachite green method. This sequence was repeated until P in the wash was below 0.1 ppm. The P in the leachate and washes was summed together to give CAP, which was then normalised to the Ca and Mg concentrations in the leachate. Three Ediacaran dolomite sediments with varying TIC, total organic carbon and clay contents were run alongside unknowns to check reproducibility for quality assurance, with duplicate CAP measurements differing by <10%.

## Biogeochemical modelling
A four-box biogeochemical model after Slomp and Cappellen (2007)[70] and Alcott et al.[61] was used to model biogeochemical feedbacks among the P, C and O₂ cycles during the GOE. The model has previously been used to explore a long-term shift in Earth's redox state, driven by the cooling mantle and the emergence and carbon enrichment of the continental crust[61]. This model is solved in MATLAB using the Ordinary Differential Equation suite. Here, we provide a key description of our model work.

In this model the ocean is divided into proximal, distal, surface and deep ocean reservoirs. The proximal zone comprises the part of the coastal ocean directly influenced by river input and includes large bays, the open-water part of estuaries, deltas, inland seas and coastal sabkhas. The distal zone is where element recycling and upwelling are quantitatively more important than river input, and which includes the open-continental shelves (average water depth is -130 m). The remaining two boxes include the surface layer of the open ocean (average depth 150 m) and the deep ocean. In the model setup from Alcott et al.[66,43], a number of forcings are applied to guide the secular rise of atmospheric $O_2$. The model uses these forcings to progressively increase the delivery of riverine P to the oceans, and/or decrease the reducing gas flux from Earth's interior to surface; both directly translate to increasingly higher atmospheric $O_2$ level over time. For our study, we induce an atmospheric $O_2$ level below $10^{-5}$ PAL using a pre-GOE reduced gas flux (Alcott et al.[43,52]) before the LE (2220 to 2060 Myr ago)[5]. A riverine P input forcing is then applied to create a rise in atmospheric $O_2$ level during the LE that is consistent with geochemical redox proxies for atmospheric $O_2$[5,71]. Some previous use of this model has included an Fe scavenging flux following prior assertion that Fe-oxides may have removed P from Precambrian oceans. This flux adds a redox-dependent negative flux to the marine P pool. We did not apply this Fe scavenging flux in our model run because it is not used in the most recent model version, which has demonstrated a reasonable fit to long-term C and P cycle data. However, as we force the model by adding P to the global ocean and do not rely on redox changes as the driver of P availability in the model, consideration of this flux would not change our conclusions.

### Calculation of proximal and distal ocean inorganic carbon isotope values

The biogeochemical model used here includes an inorganic carbon cycle with the inclusion of a one-box inorganic carbon reservoir. This allows the computation of the average isotopic composition of DIC in the oceans (see Alcott et al.[43] for further details). This simplified one-box component, however, cannot capture the heterogeneity of the inorganic carbon cycle within the oceans, which we will briefly describe below.

In the modern ocean, the C-isotopic composition of DIC ($\delta^{13}C_{DIC}$) is modulated by the effects of photosynthesis and respiration[63,72]. In the shallow portions of the ocean (<100 m depth), rates of photosynthesis outweigh those of respiration, resulting in the production of high $\delta^{13}C_{DIC}$ values. At progressively greater depths, rates of respiration increase over those of photosynthesis, resulting in lower $\delta^{13}C_{DIC}$ values.

In order to capture the heterogeneity of the ocean inorganic carbon cycle, we follow the assumption that seawater $\delta^{13}C_{DIC}$ values are related to seawater P concentrations, as is the case in the modern ocean[63]. This assumption is based on the fact that photosynthesis in shallow waters results in the uptake of P and hence lower P concentrations in shallow waters. At depth, greater rates of respiration result in the concomitant release of P and isotopically depleted C to the oceans. This innate connection between dissolved inorganic C and P results in what is referred to as the 'Redfield slope'[63]. Consequently, assuming that seawater, inorganic P and C concentrations are mainly controlled by photosynthesis and respiration, the $\delta^{13}C_{DIC}$ can be related to seawater phosphate concentration as follows[63]:

$$\delta^{13}C_{DIC} = \delta^{13}C^{A.O.} + \frac{\Delta photo}{\Sigma CO_2^{A.O.}} \frac{C_{org}}{P_{org}} (PO_4 - PO_4^{A.O.}) \tag{1}$$

In this equation, A.O. denotes average ocean. Therefore $\delta^{13}C^{A.O.}$ in our model refers to the global seawater $\delta^{13}C_{DIC}$ value (termed Aiso in the model code). $\Delta photo$ is the isotopic offset between organic and inorganic carbon, $\Sigma CO_2^{A.O.}$ corresponds to the average seawater DIC concentration, $C_{org}/P_{org}$ is the ratio of C/P in biomass, $PO_4$ is the

concentration of P in the ocean region for which $\delta^{13}C_{DIC}$ is being calculated for and $PO_4^{A.O.}$ is the average seawater P concentration, which in our model corresponds to the deep-ocean P concentration.

### Monte Carlo parameters

As shown by the sensitivity areas (green shading) in Fig. 6, several parameters were varied across 1000 model runs for a Monte Carlo analysis. In our model runs, we varied the C/P ratio of biomass, the mean seawater DIC concentration, and the magnitude of P input to the ocean.

The C/P of biomass is known to vary with the abundance of nutrients in water bodies[73]. The ratio is constrained between the Redfield ratio (106) and previously speculated values of up to 400[74].

The DIC concentration in the Paleoproterozoic oceans is uncertain, but current best estimates suggest that <30 mmol/L is most likely[75]. With this in mind, we varied the parameter between the modern seawater value of 2.2 mmol/L and 30 mmol/L as estimated by geochemical data for the Paleoproterozoic[75].

We varied the magnitude of the additional P input in our model to explore how an order of magnitude increase in P input between $10^9$ and $10^{10}$ mol P/yr would affect atmospheric $O_2$ level.

For the full model description, including model equations and parameters, see Alcott et al.[43].

## Data availability

The geochemical data generated and used in this study have been deposited in the Figshare database at https://doi.org/10.6084/m9.figshare.29380727.

## Code availability

Model code can be downloaded at GitHub (https://github.com/lalcott/d13Ctemp_2023).

## Materials availability

All samples were collected and exported in a responsible manner and in accordance with relevant permits and local laws. Global coordinates and/ or location information and drill core names are given for all samples collected in the supplementary information files. Requests for materials should be addressed to M.S.D., C.L., A.B.

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

## Acknowledgements

We thank the Geological Survey of Western Australia and Department of Mines, Industry Regulation and Safety for generously providing samples. C.L. acknowledges support from the NSFC (grants # 42130208 and 42425002). M.S.D. acknowledges support from the Forrest Research Foundation, UWA School of Earth Sciences, the International Post-doctoral Exchange Program of China, the China Postdoctoral Science Foundation and Chengdu University of Technology. M.S.D. and A.Y.S. acknowledge the Research Facility at the Centre for Microscopy, Characterisation and Analysis and Microscopy Australia Node at The University of Western Australia, funded by the University, State and Commonwealth Governments. F.O.O. thanks support from Khalifa University of Science and Technology (KUST grant FSU-2023-020/8474000494) and its Polar Research Center. A.B. acknowledges support from the NSERC Discovery and Accelerator grants and the Petroleum Foundation of the American Chemical Society (grant # 624840ND2). B.J.W.M. is supported by UKRI (grant # EP/Y008790/1). F.O.O. acknowledges support from Khalifa University of Science and Technology (KUST grant # FSU-2023-020/8474000494).

## Author contributions

M.S.D. and C.L. designed the research. M.S.D., Z.Z., H.G. and A.Y.S. performed analyses. M.S.D., B.J.W.M. and L.J.A. performed modelling. M.S.D., A.B., C.L., C.A.R. and F.P. provided samples. The paper was written by M.S.D. with contributions from C.L., A.B., C.A.R., B.J.W.M., L.J.A., F.O.O. and M.H.

## Competing interests

The authors declare no competing interests.
