## [Transparent Peer Review file · Nature Communications]

Marine phosphorus and atmospheric oxygen were coupled during the Great Oxidation Event

Corresponding Author: Dr Matthew Dodd

Version 0:

Reviewer comments:

Reviewer #1

(Remarks to the Author)

General Comments

In this paper, Dodd and coauthors present interesting data showing that the abundance of carbonate-associated phosphate is higher in rocks preserving the Lomagundi Carbon isotope excursion (LE). They claim that this is evidence that marine phosphate levels were higher during the LE, and consequently argue for a mechanistic link between P delivery to the ocean and the oxygenation of the atmosphere at the time.

I do think the data are interesting, and the idea is an enticing one. However, there are several issues with the manuscript that decrease my confidence in the conclusions drawn by the authors:

- 1) The authors too often make strong assertions without literature support and confuse interpretations with facts.
- 2) The authors offer a simplistic view of the sulfur cycle that is outdated with respect to the literature.
- 3) The authors gloss over important recent literature demonstrating the facies-dependence of the LE, which has major implications for how it might be interpreted with respect to oxygenation.
- 4) Throughout, there is a missing detailed analysis of how early stages of diagenesis might influence the main data discussed in the paper (CAP and $\delta^{13}\text{C}$ -carb).
- 5) A major and obvious alternative explanation for both elevated CAP and $\delta^{13}\text{C}$ values (evaporative conditions) is ruled out very quickly by the authors, without much justification.

I have outlined these concerns and others in detailed comments below, which I hope will be of use to the authors. I believe this work will be suitable for publication in this journal after major revisions.

Thank you for the opportunity to review this interesting work.
Roger Bryant

Line-by-line comments

Line 30: "Great Oxidation Episode" does not correspond to most usage of "GOE" in the community. Suggest the authors stick to Event.

Line 34: I'm not sure there is such a thing as a 'direct' proxy. In this case, phosphate is measured but as a proxy for oceanic phosphorus (not orthophosphate?). That seems like quite a complex, certainly indirect proxy.

Line 37: Why a phosphorus control on the global carbon cycle and not the other way around? The authors don't need me to remind them that correlation need not imply causation.

Line 50: See comment on Line 30.

Line 60-61: I think the community agrees that the GOE was protracted, but the O₂ fluctuations and oscillations are less well-agreed upon, due in complexities in proxy behavior (e.g., the sensitivity of metal speciation to both depositional and post-depositional effects). Nevertheless, if this statement represents the authors' view, that can be stated explicitly rather than presuming to speak for the broader community.

Line 62-64: "Oscillating seawater sulfate level" is an interpretation, not a fact. Here, and throughout, more careful language is needed to distinguish between observations and interpretations. In this case, there are numerous reasons why the

interpretation of oscillating seawater sulfate level might be an over-interpretation. To give one example, in reference 10 (Planavsky et al., 2012), the sulfur isotopic composition sedimentary sulfate is a piece of evidence used to argue for changes in the seawater sulfate concentration. However, one can imagine several scenarios by which the primary $\delta^{34}\text{S}$ of marine sulfate could change in the absence of changes in the size of the marine sulfate reservoir. Then, there is the issue of diagenetic alteration of carbonate-associated sulfate, which has been argued to afflict much younger rocks (see, e.g., Present et al., 2019; Murray et al., 2021), but has yet to be evaluated for most Paleoproterozoic rocks, with some recent exceptions (Bryant et al., 2024).

T. M. Present, M. Gutierrez, G. Paris, C. Kerans, J. P. Grotzinger, J. F. Adkins, Diagenetic controls on the isotopic composition of carbonate-associated sulphate in the Permian Capitan Reef Complex, West Texas. *Sedimentology*. 66, 2605–2626 (2019).

S. T. Murray, J. A. Higgins, C. Holmden, C. Lu, P. K. Swart, Geochemical fingerprints of dolomitization in Bahamian carbonates: Evidence from sulphur, calcium, magnesium and clumped isotopes. *Sedimentology*. 68, 1–29 (2021).

R. N. Bryant, J. P. Todes, J. A. Richardson, T. C. Kalia, A. R. Prave, A. Lepland, K. Kirsimäe, C. L. Blättler, Local sedimentary effects shaped key sulfur records after the Great Oxidation Event. *Earth Planet. Sci. Lett.* 648, 119113 (2024).

Lines 70-71: It seems strange to mention only the canonical interpretation of the LE here, despite recent work highlighting the facies-dependence of the magnitude of the LE (Prave et al., 2022; Hodgskiss et al., 2023), which clearly has implications for how the LE is interpreted (i.e., the LE might have required less oxidation/oxygenation than canonically assumed).

Lines 93-94: Yes, but the proportionality for a given element is often a function of the concentrations of other things in solution (e.g., Ca, Mg, carbonate ions), kinetics, etc. See, e.g., Barkan et al. (2020). One could decrease CAP for the same orthophosphate concentration by increasing the activity of CO_3^{2-} in solution.

Y. Barkan, G. Paris, S. M. Webb, J. F. Adkins, I. Halevy, Sulfur isotope fractionation between aqueous and carbonate-associated sulfate in abiotic calcite and aragonite. *Geochim. Cosmochim. Acta*. 280, 317–339 (2020).

Lines 94-95: P, or phosphate? Also, why 'relative concentration of P'?

Lines 99-101: This is quite the statement – needs literature support or further justification. To make a rough analogy with CAS, the concentration of a non-stoichiometric element in carbonate can change after burial and need not bear 'direct' relevance to the concentration of that element in the primary fluid.

Lines 119-121: This sentence is confusingly structured – consider revising for clarity.

Lines 123-138: The results section strikes me as being very short. One thing I would like to see here as a reader is some actual CAP abundances in the text, in addition to the comparisons with bracketing carbonates. If CAP works how it is supposed to, the relative changes in CAP should be less meaningful than the absolute changes. I am also wondering how one chooses how much of the bracketing carbonates to include in this relative CAP assessment.

Line 141: This should be in the results rather than the discussion, in my opinion. The analysis in the following clause can stay in the discussion.

Line 143: Here and throughout, I think the use of 'CAP values' rather than 'CAP abundance' is perhaps unhelpful. Nowhere in the text is CAP assigned a unit. One is left with a sense of CAP values being an abstract statistical measure rather than a measured quantity.

Lines 143-147: This is a lengthy sentence and complex argument. I had to read this several times to grasp the intention. I would suggest splitting the sentence up and laying out the argument more systematically without skipping logical steps.

Line 159: I think the authors have done a good job of alleviating fears about metamorphism influencing the data in a meaningful way. However, they allude here to other 'post-depositional processes'. Missing in the text is a discussion of these other post-depositional processes and how those might affect CAP abundance. Reading between the lines, the processes in question could be early marine diagenesis of carbonates. How might one screen for the influence of such processes? What about meteoric diagenesis? I would be worried about both styles of diagenesis, but particularly the latter, because it could conceivably decrease both CAP abundance and $\delta^{13}\text{C}$ -carb. Can the authors demonstrate that the bracketing units (which they say are the same sabkha facies as the LE carbonates) are not simply more influenced by meteoric diagenesis than those in the LE carbonates? Does mineralogy have a role in the preservation of CAP?

Line 163: I disagree with diagenetic and metamorphic alteration being lumped together here. The authors have demonstrated that metamorphic grades did not drive the signals they report. They have not conducted a similar analysis for any style or phase of diagenesis (although I see some work to this end further into the document).

Lines 165-167: This information should be in the results section.

Line 178: I appreciate this analysis by the authors. I would also be interested to know if there are correlations between CAP and CAS content. There are several ways one could interpret such a correlation. The first would be high orthophosphate and sulfate concentrations in the water column. High alkalinity and/or carbonate ion activity would be another.

Lines 188-189: This information should be in the results. Here, it would be useful to explain why these diagenetic indices were chosen, and what they specifically help to screen for. To my mind, Mg/Ca and Mn/Sr are necessary, but not sufficient to rule out the influence of some types of diagenesis.

Lines 194-196: This section needs references. My view is that the effect of the diurnal cycle on CAP need not necessarily be as simple as orthophosphate concentrations being drawn down throughout the water column. What if the locations of carbonate formation and photosynthesis were decoupled?

Lines 197-201: This section is extremely important but is glossed over rather quickly. These evaporitic conditions could be driving many of the signals presented in this paper. The existence of one LE-bearing Formation with a putative deeper water origin does not in my opinion make evaporative conditions an unlikely mechanism for the coupling between CAP and $\delta^{13}\text{C}$. Bear in mind also that elevated $\delta^{13}\text{C}$ values in deeper water settings are uncommon during the LE (Prave et al., 2022; Hodgskiss et al., 2023), and in some cases have been linked to remobilization and transport of shelf sediments, following Walther's Law. The authors need to do more here to justify this argument.

Line 205: Why pH and not alkalinity or carbonate ion concentration?

Lines 206-208: This sentence needs a reference. Also, what direction would pH have needed to shift?

Lines 209-211: This sentence is an oversimplification. Ruling out pH changes is not sufficient to argue that background ocean chemistry (which includes many parameters other than pH) was not an overarching control.

Line 226: This information should be in the results section.

Lines 221-227: This sentence is far too long and should be broken up for clarity.

Line 260: Given that the CIE is facies-dependent and is preserved mostly in shallow water settings, it is not necessary to make the entire ocean DIC pool 8 per mil heavier, as I assume was done in this modeling exercise.

Lines 281-287: Okay this section resolves my previous comment.

Lines 326-329: These lines are too strong for my liking. In my opinion, the data presented in this paper, while interesting, do not conclusively demonstrate that marine phosphate levels were higher during the LE, let alone that there was a mechanistic linkage between high phosphate levels and oxygenation. There is a disconnect between the complicated picture painted in the discussion and the certainty with which the conclusions are delivered.

Figure 1, panel B: This panel is very out of date, and most egregiously ignores sulfate evaporite and CAS data from the Onega Basin. See, e.g., Blättler et al. (2018). This is an important omission because there is strong evidence from that paper that seawater sulfate d34S was around 5-6 per mil right before the end of the LE.

C. L. Blättler, M. W. Claire, A. R. Prave, K. Kirsimäe, J. A. Higgins, P. V. Medvedev, A. E. Romashkin, D. V. Rychanchik, A. L. Zerkle, K. Paiste, T. Kreitsmann, I. L. Millar, J. A. Hayles, H. Bao, A. V. Turchyn, M. R. Warke, A. Lepland, Two-billion-year-old evaporites capture Earth's great oxidation. *Science*. 360, 320–323 (2018).

Reviewer #2

(Remarks to the Author)

With interest I have read “Increased phosphorus bioavailability triggered Earth's Permanent Surface Oxygenation”, in which the authors explore the role of P bioavailability during the Great Oxidation Episode (GOE), ~2,000 Ma). Across ancient sediments with varying degrees of diagenetic and metamorphic alteration, the authors present a positive correlation between a specific sedimentary P phase that is assumed to reflect the ambient seawater PO₄ concentration (carbonate-associated P) and the $\delta^{13}\text{C}_{\text{CARB}}$, for which positive excursions may reflect increased marine primary productivity (PP) which sequesters light $\delta^{13}\text{C}_{\text{CARB}}$ in organic matter. This is interpreted as evidence that PO₄ bioavailability controlled marine PP, i.e. oxygenic photosynthesis, and thereby drove oxygenation of Earth's atmosphere. A coupled biogeochemical model is subsequently used to explore the feedbacks between P input, carbon cycling and atmospheric O₂. While I find the presented data interesting, I think the storyline needs further development. Here, I also notice that little attention is given by the authors to previous studies (some I mention below) that have attempted to reconstruct ancient oceanic P cycling. In my opinion, a broader consideration of ancient ocean chemistry and P cycling would strongly benefit this submission, which is too narrowly focused on one correlation at the moment. On a personal note, from experience, chemical methods to determine P species are always uncertain, so actual ‘direct’ evidence of P retention mechanisms and CAP would be a big step forward in this field.

Paleoceanographers have to on nuanced lines of evidence to reconstruct an ancient, unknown world. This is especially true when trying to reconstruct P cycling and Earth oxygenation some 2 billion years ago. Careful conclusions have to be drawn from limited data, further complicated by long-term alteration of available sediment records. Such work can fuel new, contestable insights into the evolution of the global P cycle; recent(ish) examples are Reinhard et al., 2017 (*Nature*, 2016) Canfield et al., 2020 (*Earth-Science Reviews*), two papers that are not mentioned in this manuscript, to my surprise. The manuscript by Dodd and others that lies before me, leaves me excited yet unfulfilled. I will explain why I find the manuscript interesting but not fully developed below.

My assessment is that the authors provide some interesting data regarding P cycling around the GOE, but fail to provide an adequate framework for interpretation of the data. A recently developed proxy for seawater PO₄ (carbonate-associated P, CAP) is used well beyond the time frame for which it was “validated” (Dodd et al., 2021, *GCA*), subsequently a correlation between two proxies ($\delta^{13}\text{C}$ and CAP) is presented as a direct (this word is used again and again; a proxy is inherently indirect and correlation is not causation) evidence that the GOE was driven by increased P bioavailability. I am not convinced that a P-driven increase in PP is the only way to increase CAP; there is so little attention to carbonate formation mechanisms and P incorporation. As it stands, I could not discount that an increase in P fluxes to the sediment by any mechanism could result in an increase in P incorporation into carbonate, particularly because there is no other data presented whatsoever to illustrate the diverse depositional settings (other than shallow or not). The overselling of the data does not sit well with me; it also contrasts starkly with the consideration of numerous complicating factors regarding data interpretation that the authors mention themselves (e.g. interpretation of the $\delta^{13}\text{C}$ excursion, impact of long-term sediment alteration on CAP).

Overall:

- Proxies are not direct evidence and correlation is not causation; the $\delta^{13}\text{C}$ -CAP correlation is interesting and suggests links/feedbacks between PO₄ availability/burial and primary productivity (that have been established in numerous previous experimental and modelling studies on modern and ancient sediment), however more caution in the wording, specifically proclaiming direct evidence of P as driver of the GOE, is advisable.
- The caption to Fig. 3 states that “the high $\delta^{13}\text{C}_{\text{carb}}$ values are generally accompanied by the high CAP values” and that is more or less the extent of the evidence provided for far-reaching conclusions about drivers of the GOE. That is an interesting starting point but rather meager to build the complete story on. Using a coupled biogeochemical model in which primary productivity is controlled by PO₄ and increasing PO₄ obviously will return the desired result, but fails to provide much insight into the cascade of drivers and effects and feedbacks in the very different ocean 2 billion years ago.
- The study fails to provide a useful context for the results; there are no other data (total P, TOC, redox-sensitive elements, ...) that illustrate P cycling and the depositional settings and evolution thereof in the investigated sections. The ancient ocean may have had very specific chemistry that would impact P availability (e.g. Fe-P coupling), none of this is considered in the paper. The model apparently was parametrized with modern-ocean fluxes? It is difficult from the information in the manuscript to assess the model's suitability to explore coupled biogeochemical cycling in the ocean 2 billion years ago.
- CAP is used as direct (?) measure of seawater PO₄ concentration based on a previous study that ‘validated’ (with various

caveats) the use of this P fraction as proxy for dissolved PO₄ during formation of carbonates for sediments much younger than those investigated here. I appreciate the authors' focus on the broad co-occurrence of elevated $\delta^{13}\text{C}$ and CAP, but some discussion on the suitability of CAP in these ancient sediments, linked to mechanisms of carbonate formation, would be useful.

• It would be very interesting if the large scatter in Fig. 3, perhaps indicative of other processes acting upon CAP?) was discussed more seriously, now the reader is pushed in a certain direction with thick grey lines (what is "locally weighted scatter plot smoothing?") that brush aside what seems to me high uncertainty introduced by large scatter in low-resolution sediment records. I understand that there are statistically significant correlations presented in Fig. 2, but the scatter is large and because there is no other data (besides S isotopes to support timing of oxygenation) it is just difficult to appreciate what the plots are really telling us, and particularly difficult to conclude from this that direct evidence of P availability as driver of the GOE is provided.

Some specific comments:

L34-35. You are not reconstructing oceanic PO₄ concentrations (this suggests that you provide information on concentrations), you are qualitatively evaluating whether the sedimentary CAP record shows evidence for variations in seawater PO₄ concentrations.

L41. Is the positive correlation between CAP and $\delta^{13}\text{C}$ direct evidence that PO₄ availability drove the GOE? Isn't correlation between two proxies by definition indirect evidence?

L50. The time bracket indicated for the GOE in Fig. 1 actually succeeds the initial permanent rise, starting at the first peak.

L51-52. As the GOA was the oxygenation of the atmosphere, "paved the way" seems awkward phrasing because it implies another subsequent process was responsible for that.

L59. Indicate collapse in Figure and perhaps add indication of shift from 10-1 to 10-3 PAL

L60. In Figure 1, it's Great Oxygenation Event.

L71. "and consequently net oxidation of the surface environment" may require some additional clarification for the broad readership.

L88-89. A proxy is by definition indirect, now it's being sold almost as a PO₄ sensor.

L92. A proxy is not direct.

L102. Validated for sediments up to Ediacaran (630-530 Ma), not 2,000 Ma

L110-11. Well, whether changes in P availability may have occurred during the GOE

L159-161. This is the actual observation. From this to 'direct evidence that P availability is the direct driver of the GOE' is a long way for me.

L165-167. Also valid for these billions-years-old rocks?

L207-210. Please elaborate a bit on the required pH shift. How big could the pH shift be?

L239. Please explain the feedbacks a bit more.

L255-258. The model is parametrized for the modern ocean?

L311-312. I think more insight into the global, 'full-ocean' nature of this event would be very useful; global increase in ocean primary productivity is needed to drive the isotopic event?

Reviewer #3

(Remarks to the Author)

Version 1:

Reviewer comments:

Reviewer #1

(Remarks to the Author)

I commend the authors on doing a good job responding to my earlier concerns, and I congratulate them on a thought-provoking paper. It will be interesting to see how work on the CAP proxy and the GOE evolves in the coming years.

Reviewer #2

(Remarks to the Author)

The authors took care to incorporate the comments from 3 reviewers into this revised version of the manuscript. With the altered wording, the authors to my opinion mostly addressed the main criticism of all three reviewers, namely the caution required when using a sedimentary proxy to reconstruct ancient seawater.

Of course, in paleoceanography large uncertainties inherently exist and all three reviewers agreed that Dodd et al. provide valuable new data that help shed further light on the co-evolution of the marine P cycle and atmospheric oxygenation.

In the revised discussion, Dodd et al. now include existing work regarding ancient marine P cycling (e.g. Reinhard et al., Canfield et al.) and paint a more nuanced picture of their results, which I appreciate. What I do note that is that the new level of caution in interpretation of results is not included in the title; I understand the wish for a snappy title, but I would recommend a less definitive statement, considering the amount of caution now expressed in the manuscript.

Regarding the link between CAP and the size of the oceanic dissolved P reservoir, I remain of the opinion that additional data such as TOC, total P, trace metals would contribute significantly to our understanding of the sedimentary and depositional context (and P cycling) that helps shape the CAP record as well. The authors consider this 'beyond the scope of this study' and it likely complicates the story, but I find it a missed opportunity to more fully develop our understanding of the P cycle at that time. Furthermore, the authors mention that "This study provides evidence for how ocean P level was changing, but not what was causing it to change" but my point is more that additional data would help better understand sedimentary P cycles that helps shape the CAP record. That being said, also considering the feedback from the other reviewers, I appreciate that Dodd et al. included more elaborate discussion on ancient P cycling from other studies.

For the modelling, appreciate that in the revised manuscript, Dodd et al. emphasize the evidence that suggest a "P-driven GOE" (C isotopic record, pO₂ reconstructions).

Overall, Dodd et al. provide a revised manuscript that includes a more nuanced and therefore stronger presentation and discussion of the data.

A minor point in the revised manuscript with tracked changes is that the phrasing in L789-791 is a bit confusing: "Rising sulphate levels would have stimulated microbial sulphate reduction and enhanced P recycling from organic rich sediments. P speciation data from this interval support increased recycling efficiency under more oxidizing conditions due to sedimentary sulphur cycling"

This feels counterintuitive; P recycling is generally more efficient under more reducing sedimentary conditions, in this case more sulfide generation and subsequent trapping of Fe in sulphides. I think the authors refer to a more oxygenated atmosphere that would result in more sulfate supply to the oceans? It's probably useful to clarify these aspects a bit (both how enhanced sulphate reduction affects P burial efficiency and what part of the atmosphere-ocean water-seafloor system is more oxygenated).

Reviewer #3

(Remarks to the Author)

I recommend publication of the revised version of this manuscript.

Note: all line numbers refer to manuscript file without tracked changes

Reviewer #1 (Remarks to the Author):

General Comments

In this paper, Dodd and coauthors present interesting data showing that the abundance of carbonate-associated phosphate is higher in rocks preserving the Lomagundi Carbon isotope excursion (LE). They claim that this is evidence that marine phosphate levels were higher during the LE, and consequently argue for a mechanistic link between P delivery to the ocean and the oxygenation of the atmosphere at the time.

I do think the data are interesting, and the idea is an enticing one. However, there are several issues with the manuscript that decrease my confidence in the conclusions drawn by the authors:

- 1) The authors too often make strong assertions without literature support and confuse interpretations with facts.
- 2) The authors offer a simplistic view of the sulfur cycle that is outdated with respect to the literature.
- 3) The authors gloss over important recent literature demonstrating the facies-dependence of the LE, which has major implications for how it might be interpreted with respect to oxygenation.
- 4) Throughout, there is a missing detailed analysis of how early stages of diagenesis might influence the main data discussed in the paper (CAP and d13C-carb).
- 5) A major and obvious alternative explanation for both elevated CAP and d13C values (evaporative conditions) is ruled out very quickly by the authors, without much justification.

I have outlined these concerns and others in detailed comments below, which I hope will be of use to the authors. I believe this work will be suitable for publication in this journal after major revisions.

Thank you for the opportunity to review this interesting work.
Roger Bryant

1.11

We thank the reviewer for their support of this work and time spent and comments offered. Our detailed responses to the major comments raised by Dr. Bryant above are provided below in our responses to related lined comments.

Line-by-line comments

Line 30: "Great Oxidation Episode" does not correspond to most usage of "GOE" in the community. Suggest the authors stick to Event.

1.12

Changed to Great Oxidation Event

Line 34: I'm not sure there is such a thing as a 'direct' proxy. In this case, phosphate is measured but as a proxy for oceanic phosphorus (not orthophosphate?). That seems like quite a complex, certainly indirect proxy.

1.13

We agree that all proxies are indirect and have removed the wording 'direct' proxy.

Line 37: Why a phosphorus control on the global carbon cycle and not the other way around? The authors don't need me to remind them that correlation need not imply causation.

1.14

We acknowledge this important consideration which was also raised by the other reviewers. We have expanded on this in lines 318-329 and have tempered the language in the abstract.

Line 50: See comment on Line 30.

1.15

Changed wording.

Line 60-61: I think the community agrees that the GOE was protracted, but the O₂ fluctuations and oscillations are less well-agreed upon, due in complexities in proxy behavior (e.g., the sensitivity of metal speciation to both depositional and post-depositional effects). Nevertheless, if this statement represents the authors' view, that can be stated explicitly rather than presuming to speak for the broader community.

1.16

Agreed and has been reworded to reflect this.

Line 62-64: "Oscillating seawater sulfate level" is an interpretation, not a fact. Here, and throughout, more careful language is needed to distinguish between observations and interpretations. In this case, there are numerous reasons why the interpretation of oscillating seawater sulfate level might be an over-interpretation. To give one example, in reference 10 (Planavsky et al., 2012), the sulfur isotopic composition sedimentary sulfate is a piece of evidence used to argue for changes in the seawater sulfate concentration. However, one can imagine several scenarios by which the primary $\delta^{34}\text{S}$ of marine sulfate could change in the absence of changes in the size of the marine sulfate reservoir. Then, there is the issue of diagenetic alteration of carbonate-associated sulfate, which has been argued to afflict much younger rocks (see, e.g., Present et al., 2019; Murray et al., 2021), but has yet to be evaluated for most Paleoproterozoic rocks, with some recent exceptions (Bryant et al., 2024).

1.17

We have adjusted the manuscript wording to clearly distinguish between observational data and the interpretation of "oscillating seawater sulfate levels." We further acknowledge possible scenarios where $\delta^{34}\text{S}$ changes might not necessarily reflect fluctuations in sulfate reservoir size, citing additional literature highlighting these complexities.

T. M. Present, M. Gutierrez, G. Paris, C. Kerans, J. P. Grotzinger, J. F. Adkins, Diagenetic controls on the isotopic composition of carbonate-associated sulphate in the Permian Capitan Reef Complex, West Texas. *Sedimentology*. 66, 2605–2626 (2019).

S. T. Murray, J. A. Higgins, C. Holmden, C. Lu, P. K. Swart, Geochemical fingerprints of dolomitization in Bahamian carbonates: Evidence from sulphur, calcium, magnesium and clumped isotopes. *Sedimentology*. 68, 1–29 (2021).

R. N. Bryant, J. P. Todes, J. A. Richardson, T. C. Kalia, A. R. Prave, A. Lepland, K. Kirsimäe, C. L. Blättler, Local sedimentary effects shaped key sulfur records after the Great Oxidation Event. *Earth Planet. Sci. Lett.* 648, 119113 (2024).

Lines 70-71: It seems strange to mention only the canonical interpretation of the LE here, despite recent work highlighting the facies-dependence of the magnitude of the LE (Prave et al., 2022; Hodgskiss et al., 2023), which clearly has implications for how the LE is interpreted (i.e., the LE might have required less oxidation/oxygenation than canonically assumed).

1.18

We have acknowledged other interpretations of the Lomagundi Excursion on lines 80-82. We also show in our biogeochemical model that a facies-dependent expression of the LE aligns with our data and conclusions if the Paleoproterozoic carbon gradient varied with depth like in the modern ocean, see lines 365-368.

Lines 93-94: Yes, but the proportionality for a given element is often a function of the concentrations of other things in solution (e.g., Ca, Mg, carbonate ions), kinetics, etc. See, e.g., Barkan et al. (2020). One could decrease CAP for the same orthophosphate concentration by increasing the activity of CO₃²⁻ in solution.

Y. Barkan, G. Paris, S. M. Webb, J. F. Adkins, I. Halevy, Sulfur isotope fractionation between aqueous and carbonate-associated sulfate in abiotic calcite and aragonite. *Geochim. Cosmochim. Acta.* 280, 317–339 (2020).

1.19

Yes, this is true, we have clarified that phosphate incorporation into carbonate minerals is influenced by multiple environmental parameters including ion concentrations and kinetics on lines 92-94 and lines 209-230 and acknowledge this as a limitation.

Lines 94-95: P, or phosphate? Also, why 'relative concentration of P'?

1.20

We have now clarified P as phosphate and removed the word "relative".

Lines 99-101: This is quite the statement – needs literature support or further justification. To make a rough analogy with CAS, the concentration of a non-stoichiometric element in carbonate can change after burial and need not bear 'direct' relevance to the concentration of that element in the primary fluid.

1.21

We have reworded to "...while phosphate bound inside carbonate minerals can be attributable to ambient dissolved P concentrations."

Lines 119-121: This sentence is confusingly structured – consider revising for clarity.

1.22

We have reworded on lines 118-121

Lines 123-138: The results section strikes me as being very short. One thing I would like to see here as a reader is some actual CAP abundances in the text, in addition to the comparisons with bracketing carbonates. If CAP works how it is supposed to, the relative changes in CAP should be less meaningful than the absolute changes. I am also wondering how one chooses how much of the bracketing carbonates to include in this relative CAP assessment.

1.23

We have added CAP values to show the range of absolute values on lines 134-135 and 143-144. We used a cut-off of 3 per mil to compare CAP values across carbon isotope excursions, we do not expect this cut-off to play a major factor in the comparison as Figure 2 shows the correlation of CAP and d13C is continuous and not a phenomenon developing from discrete data binning. Regarding relative vs. absolute change it is important to note that absolute values could range between localities if some ocean regions had higher local phosphate availability as discussed on lines 232-234 or varying degrees of metamorphism lower CAP values. Take the Nash fork location for example, it shows a change in absolute CAP values across the LE, but if you compare these absolute values to the Francevillian section they are considerably lower. Therefore, we use relative to describe the fact that despite differences in absolute values between sections they show the same trends relative to the CAP values from the individual sections.

Line 141: This should be in the results rather than the discussion, in my opinion. The analysis in the following clause can stay in the discussion.

1.24

Yes, this sentence contains both a result descriptor but also an interpretive clause. We have chosen to leave it as a bridge between sections.

Line 143: Here and throughout, I think the use of 'CAP values' rather than 'CAP abundance' is perhaps unhelpful. Nowhere in the text is CAP assigned a unit. One is left with a sense of CAP values being an abstract statistical measure rather than a measured quantity.

1.25

We have now detailed the expression of CAP units on lines 102-104

Lines 143-147: This is a lengthy sentence and complex argument. I had to read this several times to grasp the intention. I would suggest splitting the sentence up and laying out the argument more systematically without skipping logical steps.

1.26

Agreed we have removed as it is discussed later in the text around basin preservation over time on lines 231-244.

Line 159: I think the authors have done a good job of alleviating fears about metamorphism influencing the data in a meaningful way. However, they allude here to other 'post-depositional processes'. Missing in the text is a discussion of these other post-depositional processes and how those might affect CAP abundance. Reading between the lines, the processes in question could be early marine diagenesis of carbonates. How might one screen for the influence of such processes? What about meteoric diagenesis? I would be worried about both styles of diagenesis, but particularly the latter, because it could conceivably decrease both CAP abundance and d13C-carb. Can the authors demonstrate that the bracketing units (which they say are the same sabkha facies as the LE carbonates) are not simply more influenced by meteoric diagenesis than those in the LE carbonates? Does mineralogy have a role in the preservation of CAP?

1.27

We appreciate the support of our discussion of metamorphic influences on CAP. We have moved the metamorphic discussion so it now follows the expanded discussion on diagenetic processes on lines 234-261. Yes, mineralogy does affect CAP preservation and is discussed on lines 93 and 116.

Line 163: I disagree with diagenetic and metamorphic alteration being lumped together here. The authors have demonstrated that metamorphic grades did not drive the signals they report. They have not conducted a similar analysis for any style or phase of diagenesis (although I see some work to this end further into the document).

1.28

We have expanded our discussion on diagenesis to cover a wide array of hypothetical diagenetic scenarios on lines 146-190 and include new data statistics which compare carbonate trace element chemistry and CAP to discuss diagenetic influences.

Lines 165-167: This information should be in the results section.

1.29

Agreed.

Line 178: I appreciate this analysis by the authors. I would also be interested to know if there are correlations between CAP and CAS content. There are several ways one could interpret such a correlation. The first would be high orthophosphate and sulfate concentrations in the water column. High alkalinity and/or carbonate ion activity would be another.

1.30

We did not measure CAS contents of the sample analysed for CAP, however we can use CAS values from the literature to give a broad overview. Based on the literature, CAS is generally elevated in carbonate from the LE relative to those carbonates deposited outside the LE. This would mean there would be a general positive correlation of CAP and CAS across the LE. We would favour the first interpretation put forward by the reviewer, given exceptionally high alkalinity increases would be required to explain the CAP and CAS increases on a solely alkalinity-controlled basis. For example, Barkan et al., 2018 found a roughly 1:1 relative increase in alkalinity to CAS concentration which based on data in Planavsky et al., 2012 which show a rough change from ca. 50ppm to 250ppm across the LE, which would require a ca. >5x increase in marine alkalinity to explain the CAS concentration change across the LE. This would result in an increase in ocean pH if not matched with a corresponding increase in atmospheric CO₂, which is unlikely for the GOE interval, based on existing evidence for glaciations.

Lines 188-189: This information should be in the results. Here, it would be useful to explain why these diagenetic indices were chosen, and what they specifically help to screen for. To my mind, Mg/Ca and Mn/Sr are necessary, but not sufficient to rule out the influence of some types of diagenesis.

1.31

We have now included a description of why the diagenetic indices were used and other possible diagenetic indices on lines 176-178.

Lines 194-196: This section needs references. My view is that the effect of the diurnal cycle on CAP need not necessarily be as simple as orthophosphate concentrations being drawn down throughout the water column. What if the locations of carbonate formation and photosynthesis were decoupled?

1.32

The diurnal cycle explicitly requires carbonate formation in-situ in order for carbonate to capture changing carbon isotopic composition from local photosynthetic activity

(Geyman and Maloof, 2019). Maybe future works may be able to build a more complex cycle but based on the current state of knowledge we can confidently rule this out.

Lines 197-201: This section is extremely important but is glossed over rather quickly. These evaporitic conditions could be driving many of the signals presented in this paper. The existence of one LE-bearing Formation with a putative deeper water origin does not in my opinion make evaporative conditions an unlikely mechanism for the coupling between CAP and d13C. Bear in mind also that elevated d13C values in deeper water settings are uncommon during the LE (Prave et al., 2022; Hodgskiss et al., 2023), and in some cases have been linked to remobilization and transport of shelf sediments, following Walther's Law. The authors need to do more here to justify this argument.

1.33

We have discussed this phenomenon in more detail on lines 204-208 and include new Y/Ho data to develop this point.

Line 205: Why pH and not alkalinity or carbonate ion concentration?

1.34

These effects have been previously documented in experiments and pH was found to exert the biggest influence on CAP incorporation (Dodd et al., 2023; Dodd et al., 2021). We have now noted earlier in the manuscript that alkalinity/ carbonate ion concentration also effects CAP.

Lines 206-208: This sentence needs a reference. Also, what direction would pH have needed to shift?

1.35

Reference now added on line 227 (Haley and Bachan 2017). The pH change direction does not matter unless we are specifically talking about which way the CAP values changed. Here we simply state the change whether it be higher or lower was too large to be likely explained by pH alone.

Lines 209-211: This sentence is an oversimplification. Ruling out pH changes is not sufficient to argue that background ocean chemistry (which includes many parameters other than pH) was not an overarching control.

1.36

We agree there are many chemical effects which can impact CAP incorporation into carbonate and have expanded discussion on this on lines 209-230.

Line 226: This information should be in the results section.

1.37

These lines are a discussion element with reference to the results. We have added a description of what the sample mineralogy is in the results section.

Lines 221-227: This sentence is far too long and should be broken up for clarity.

1.38

Agreed and has been modified.

Line 260: Given that the CIE is facies-dependent and is preserved mostly in shallow water settings, it is not necessary to make the entire ocean DIC pool 8 per mil heavier, as I

assume was done in this modeling exercise.

Lines 281-287: Okay this section resolves my previous comment.

Lines 326-329: These lines are too strong for my liking. In my opinion, the data presented in this paper, while interesting, do not conclusively demonstrate that marine phosphate levels were higher during the LE, let alone that there was a mechanistic linkage between high phosphate levels and oxygenation. There is a disconnect between the complicated picture painted in the discussion and the certainty with which the conclusions are delivered.

1.39

We have adopted a more conservative concluding statement in line with caveats and limitations of the CAP proxy and connection with carbon isotope values on lines 415-433

Figure 1, panel B: This panel is very out of date, and most egregiously ignores sulfate evaporite and CAS data from the Onega Basin. See, e.g., Blattler et al. (2018). This is an important omission because there is strong evidence from that paper that seawater sulfate $\delta^{34}\text{S}$ was around 5-6 per mil right before the end of the LE.

C. L. Blättler, M. W. Claire, A. R. Prave, K. Kirsimäe, J. A. Higgins, P. V. Medvedev, A. E. Romashkin, D. V. Rychanchik, A. L. Zerkle, K. Paiste, T. Kreitsmann, I. L. Millar, J. A. Hayles, H. Bao, A. V. Turchyn, M. R. Warke, A. Lepland, Two-billion-year-old evaporites capture Earth's great oxidation. *Science*. 360, 320–323 (2018).

1.40

We have added the new data from Blattler to the Fig. 1 panel B.

Reviewer #2 (Remarks to the Author):

With interest I have read “Increased phosphorus bioavailability triggered Earth’s Permanent Surface Oxygenation”, in which the authors explore the role of P bioavailability during the Great Oxidation Episode (GOE), ~2,000 Ma). Across ancient sediments with varying degrees of diagenetic and metamorphic alteration, the authors present a positive correlation between a specific sedimentary P phase that is assumed to reflect the ambient seawater PO₄ concentration (carbonate-associated P) and the $\delta^{13}\text{C}_{\text{CARB}}$, for which positive excursions may reflect increased marine primary productivity (PP) which sequesters light $\delta^{13}\text{C}_{\text{CARB}}$ in organic matter. This is interpreted as evidence that PO₄ bioavailability controlled marine PP, i.e. oxygenic photosynthesis, and thereby drove oxygenation of Earth’s atmosphere. A coupled biogeochemical model is subsequently used to explore the feedbacks between P input, carbon cycling and atmospheric O₂. While I find the presented data interesting, I think the storyline needs further development. Here, I also notice that little attention is given by the authors to previous studies (some I mention below) that have attempted to reconstruct ancient oceanic P cycling. In my opinion, a broader consideration of ancient ocean chemistry and P cycling would strongly benefit this submission, which is too narrowly focused on one correlation at the moment. On a personal note, from experience, chemical methods to determine P species are always uncertain, so actual ‘direct’ evidence of P retention mechanisms and CAP would be a big step forward in this field.

Paleoceanographers have to on nuanced lines of evidence to reconstruct an ancient, unknown world. This is especially true when trying to reconstruct P cycling and Earth oxygenation some 2 billion years ago. Careful conclusions have to be drawn from limited data, further complicated by long-term alteration of available sediment records. Such work can fuel new, contestable insights into the evolution of the global P cycle; recent(ish) examples are Reinhard et al., 2017 (Nature, 2016) Canfield et al., 2020 (Earth-Science Reviews), two papers that are not mentioned in this manuscript, to my surprise. The manuscript by Dodd and others that lies before me, leaves me excited yet unfulfilled. I will explain why I find the manuscript interesting but not fully developed below.

2.11 We thank the reviewer for their time spent and comments offered and agree the inclusion of further works pertaining to ancient P cycling can help frame the importance of this study.

My assessment is that the authors provide some interesting data regarding P cycling around the GOE, but fail to provide an adequate framework for interpretation of the data. A recently developed proxy for seawater PO₄ (carbonate-associated P, CAP) is used well beyond the time frame for which it was “validated” (Dodd et al., 2021, GCA), subsequently a correlation between two proxies ($\delta^{13}\text{C}$ and CAP) is presented as a direct (this word is used again and again; a proxy is inherently indirect and correlation is not causation) evidence that the GOE was driven by increased P bioavailability. I am not convinced that a P-driven increase in PP is the only way to increase CAP; there is so little attention to carbonate formation mechanisms and P incorporation. As it stands, I could not discount that an increase in P fluxes to the sediment by any mechanism could result in an increase in P incorporation into carbonate, particularly because there is no other data presented whatsoever to illustrate the diverse depositional settings (other than shallow or not). The overselling of the data does not sit well with me; it also contrasts starkly with the consideration of numerous complicating factors regarding data interpretation that the authors mention themselves (e.g. interpretation of the $\delta^{13}\text{C}$ excursion, impact of long-term sediment alteration on CAP).

Overall:

- Proxies are not direct evidence and correlation is not causation; the $\delta^{13}\text{C}$ -CAP correlation is interesting and suggests links/feedbacks between PO₄ availability/burial and primary

productivity (that have been established in numerous previous experimental and modelling studies on modern and ancient sediment), however more caution in the wording, specifically proclaiming direct evidence of P as driver of the GOE, is advisable.

2.12

We agree that all proxies are indirect and have removed the wording ‘direct’ proxy. We acknowledge this important consideration which was also raised by the other reviewers. We have expanded on this in lines 318-329 and have tempered the language in the abstract.

- The caption to Fig. 3 states that “the high $\delta^{13}\text{C}_{\text{carb}}$ values are generally accompanied by the high CAP values” and that is more or less the extent of the evidence provided for far-reaching conclusions about drivers of the GOE. That is an interesting starting point but rather meager to build the complete story on. Using a coupled biogeochemical model in which primary productivity is controlled by PO_4 and increasing PO_4 obviously will return the desired result, but fails to provide much insight into the cascade of drivers and effects and feedbacks in the very different ocean 2 billion years ago.

2.13

We do not see this result as “meagre”, considering that higher ocean P levels during the LE were a key testable prediction of the P-driven productivity model of the GOE. Nevertheless, we have now elaborated on the comparison of the biogeochemical model results and observations of fluctuating redox conditions during the GOE on lines 386-396. Our new model results show that if the GOE occurs because of a reduction in oxygen sinks, a stepwise transition from low to high atmospheric O_2 occurs, whereas if an increase in P-driven productivity is a driver for the GOE, atmospheric O_2 level rises and falls, which is more consistent with redox proxies.

- The study fails to provide a useful context for the results; there are no other data (total P, TOC, redox-sensitive elements, ...) that illustrate P cycling and the depositional settings and evolution thereof in the investigated sections. The ancient ocean may have had very specific chemistry that would impact P availability (e.g. Fe-P coupling), none of this is considered in the paper. The model apparently was parametrized with modern-ocean fluxes? It is difficult from the information in the manuscript to assess the model’s suitability to explore coupled biogeochemical cycling in the ocean 2 billion years ago.

2.14

Indeed, it would be interesting to have other tracers of P cycling during the GOE, however diving further into what was controlling P level during the GOE is beyond the scope of this study. This study provides evidence for how ocean P level was changing, but not what was causing it to change. We agree that further details should be added to provide a better overview of what may have been controlling ocean P level on lines 265-306.

The biogeochemical model only uses modern fluxes to recreate the present day. When it is run over Earth history (Alcott et al., 2019; Alcott et al., 2024) its fluxes are computed dynamically based on elemental abundances, climate and tectonics, and the overall model output compares reasonably well to a suite of long-term proxies. It is probably the best tested model of this type currently available. We have emphasised this now in the methods description.

- CAP is used as direct (?) measure of seawater PO_4 concentration based on a previous study that ‘validated’ (with various caveats) the use of this P fraction as proxy for dissolved PO_4 during formation of carbonates for sediments much younger than those investigated

here. I appreciate the authors' focus on the broad co-occurrence of elevated $\delta^{13}\text{C}$ and CAP, but some discussion on the suitability of CAP in these ancient sediments, linked to mechanisms of carbonate formation, would be useful.

2.15

The CAP proxy was validated principally through experimental work, which synthesised carbonates under analogous conditions to modern seawater. This abiotic precipitation style would be similar to the Precambrian world. Others have also tested CAP incorporation using analogues of Proterozoic seawater (Roest-Ellis et al., 2023) and achieve similar results. The biggest unknown would be CAP incorporation into dolomite which was a pervasive carbonate with debated origin in the Precambrian. We have added further information on lines 114-116, which detail the effects of mineralogy on CAP.

- It would be very interesting if the large scatter in Fig. 3, perhaps indicative of other processes acting upon CAP?) was discussed more seriously, now the reader is pushed in a certain direction with thick grey lines (what is "locally weighted scatter plot smoothing?) that brush aside what seems to me high uncertainty introduced by large scatter in low-resolution sediment records. I understand that there are statistically significant correlations presented in Fig. 2, but the scatter is large and because there is no other data (besides S isotopes to support timing of oxygenation) it is just difficult to appreciate what the plots are really telling us, and particularly difficult to conclude from this that direct evidence of P availability as driver of the GOE is provided.

2.16

While there is indeed significant point-to-point scatter in the CAP profiles, which is an inherent feature of low-resolution sampling in deep time carbonate archives, we argue that the observed stratigraphic trends are unlikely to be primarily diagenetic. First, we note that the CAP excursions are observed consistently across distinct basins (Francevillian and Snowy Pass), each with different diagenetic and burial histories, yet exhibiting broadly synchronous patterns when normalized to formation thickness. This stratigraphic alignment supports a common paleoenvironmental driver affecting both carbon and phosphorus systems, rather than localized diagenetic overprinting.

The 3 sections in Figure 3 record the Lomagundi and Woolly Dolomite carbon isotope excursion and show continuous chemostratigraphic trends of correlated $\delta^{13}\text{C}$ and CAP which show continuous correlation of $\delta^{13}\text{C}$ and CAP across the Lomagundi Event, as opposed to comparing individual formations from discrete time points. Given the Lomagundi Event is tied to the rise and fall in O_2 levels via other studies, the CAP trends across these sections supports their connection with higher and lower O_2 levels (Canfield et al., 2013). We have also added extra discussion on the possible impacts of post-depositional alteration as a mechanism of scatter in the data on lines 187-192.

Locally weighted scatterplot smoothing (LOWESS) is a common technique applied in stratigraphic geochemical analysis. It is a non-parametric regression technique that fits local polynomial functions to the data, allowing for visualization of broader trends while minimizing the influence of short-wavelength noise. We have now clarified this in the figure caption. The smoothing curve is included not to obscure scatter, but to guide the reader in identifying stratigraphic trajectories that are difficult to interpret in noisy datasets typical of Precambrian carbonates.

We have added the following clarifying text in the revised figure 3 caption: “Dark-grey lines represent locally weighted scatter plot smoothing”

Some specific comments:

L34-35. You are not reconstructing oceanic PO₄ concentrations (this suggests that you provide information on concentrations), you are qualitatively evaluating whether the sedimentary CAP record shows evidence for variations in seawater PO₄ concentrations.

2.17

Agreed, qualitatively we can say concentrations were potentially higher or lower without putting an absolute value to them.

L41. Is the positive correlation between CAP and d¹³C direct evidence that PO₄ availability drove the GOE? Isn't correlation between two proxies by definition indirect evidence?

2.18

We agree that despite there being evidence for temporally correlated increases in ocean P, O₂ and biomass burial we cannot conclude that P-driven biomass burial was the sole driver of the GOE. We have added this caveat on lines 320-321 and 420-444.

L50. The time bracket indicated for the GOE in Fig. 1 actually succeeds the initial permanent rise, starting at the first peak.

2.19

We have extended the arrow to capture that.

L51-52. As the GOA was the oxygenation of the atmosphere, “paved the way” seems awkward phrasing because it implies another subsequent process was responsible for that.

2.20

We have reworded to “which established a permanent oxygen atmosphere on Earth”

L59. Indicate collapse in Figure and perhaps add indication of shift from 10⁻¹ to 10⁻³ PAL

2.21

Added.

L60. In Figure 1, it's Great Oxygenation Event.

2.22

We have reworded to Event throughout the manuscript.

L71. “and consequently net oxidation of the surface environment” may require some additional clarification for the broad readership.

2.23

We have changed to “oxygenation of the surface environment.”

L88-89. A proxy is by definition indirect, now it's being sold almost as a PO₄ sensor.

2.24

We agree that all proxies are indirect and have removed the wording ‘direct’ proxy.

L92. A proxy is not direct.

2.25

We agree that all proxies are indirect and have removed the wording ‘direct’ proxy.

L102. Validated for sediments up to Ediacaran (630-530 Ma), not 2,000 Ma

2.26

The CAP proxy was validated principally through experimental work, which synthesised carbonates under analogous conditions to modern seawater. This abiotic precipitation style would be similar to the Precambrian world. Others have also tested CAP incorporation using analogues of Proterozoic seawater (Roest-Ellis et al., 2023) and achieve similar results. The biggest unknown would be CAP incorporation into dolomite which was a pervasive carbonate with debated origin in the Precambrian. We have added further information on lines 114-116, which detail the effects of mineralogy on CAP.

L110-11. Well, whether changes in P availability may have occurred during the GOE

2.27

Reworded to “...whether global seawater phosphate availability changed during the GOE”

L159-161. This is the actual observation. From this to ‘direct evidence that P availability is the direct driver of the GOE’ is a long way for me.

2.28

We agree that despite there being evidence for temporally correlated increases in ocean P, O₂ and biomass burial we cannot conclude that P-driven biomass burial was the sole driver of the GOE. We have added this caveat on lines 320-321 and 420-444.

L165-167. Also valid for these billions-years-old rocks?

2.29

The CAP proxy was validated and tested in modern and Proterozoic-like seawater (Dodd et al., 2021; Ingalls et al., 2020; Ingalls et al., 2022; Ren et al., 2025; Richardson et al., 2022; Roest-Ellis et al., 2023) and achieve similar results.

L207-210. Please elaborate a bit on the required pH shift. How big could the pH shift be?

2.30

We have now referenced a study which modelled pH changes during the GOE, with 95% of modelled results returning a pH drop of < 1 unit (Halevy and Bachan, 2017).

L239. Please explain the feedbacks a bit more.

2.31

We have added further explanation of P cycling in the Paleoproterozoic on lines 267-331

L255-258. The model is parametrized for the modern ocean?

2.32

As above, the model is developed around the modern ocean but the fluxes are allowed to vary dynamically through Earth history, so during the GOE the model is very different to the modern.

L311-312. I think more insight into the global, 'full-ocean' nature of this event would be very useful; global increase in ocean primary productivity is needed to drive the isotopic event?

2.33

We agree and have added additional discussion on lines 404-419.

Reviewer #3 (Remarks to the Author):

Review of Increased phosphorus bioavailability triggered Earth's permanent surface oxygenation by Dodd et al.

Overview: The authors present carbonate-associated phosphate (CAP) data from globally distributed marine sediments across the GOE and demonstrate a positive covariation with carbonate carbon isotopes. The primary finding of this analysis is that CAP values are high during the Lomagundi positive carbon isotope excursion. These data are used in a biogeochemical box model to support how increased P availability would lead to a rapid oxygenation of the atmosphere by increased primary productivity (photosynthesis).

Comments:

1. In several places, including a section title in the Introduction, the authors refer to CAP as a direct proxy for seawater phosphate concentrations. However, the first author's 2021 paper demonstrates a strong dependence of phosphate incorporation into carbonate minerals on pH and mineralogy. I recommend tempering the language around direct proxy if the evaluation is based on qualitative comparisons between CAP datasets across time without quantifying seawater [P]. I want to make clear that I think there is significant value in these qualitative comparisons.

3.11

We agree with tempering the language around CAP being a direct proxy and have included discussion of this on lines 33-35, 85-87, and 101-104.

2. I appreciate the approach of considering metamorphic grade in your diagenetic assessment of these samples. However, I question whether comparing across similar metamorphic grade is valid. Presumably, alteration of CAP values is more related to the phosphate contents of the diagenetic fluid rather than the P-T regime of the alteration environment (other than potentially extent of water-rock exchange). I recommend performing some amount of petrography on samples from each formation and analyzing a clear secondary fabric for CAP, such as a secondary vein. This will provide some information about the phosphate contents of altering fluids which will then provide a means of assessing whether CAP values in your fabrics interpreted as primary provide a minimum or maximum cap on CAP. A few analyses of secondary fabrics would really bolster your argument for accepting CAP values as primary, or at least usable for comparison across geographic regions.

3.12

We analysed CAP over a P-T range because prior work has shown that progressively increasing metamorphism can lead to increasing CAP loss (Ingalls et al., 2022). We agree that the degree of alteration will be unique for each case and could explain some spread in data but is highly unlikely to recreate the observed correlation in CAP and $\delta^{13}\text{C}$ across such a huge range of metamorphic grades seen in this study. Our samples were carefully selected to avoid visible grains before selecting a sample for crushing and analysis. There are however instances where we do not need to consider metamorphism to support the conclusions such as the FC Formation, Woolly Dolomite and Nash Fork Formation, which preserved continuous sections of consistent metamorphic grade and all preserve a positive correlation of CAP and $\delta^{13}\text{C}$ suggesting metamorphism is an unlikely cause of the correlation.

Petrography has been performed on these samples in prior published studies (Kalderon-Asael et al., 2021; Ossa Ossa et al., 2018). We subsampled the samples

previously used in these other studies. When subsampling we avoided visible alteration or veining.

3. Rationale for spread in CAP values: In the very first sentence of the Discussion, the authors state “There is a large spread in CAP values of formations deposited during the LE, which is expected given the heterogeneous nature of phosphorus concentration in the surface ocean.” Can the spread realistically be explained by a reasonable range of surface seawater [P]? I think it is more likely pointing to variable degrees of water-rock alteration and/or phosphate contents of altering fluids in the metamorphic environment. To be clear, I don’t think this takes away from your main finding that there is a positive covariance in CAP and d13C across the LE, but I recommend casting this within a more cautious light with respect to primacy of CAP values.

3.13

We agree with the reviewer that the results should be discussed in a more cautious light and have added this discussion throughout the text. We keep the original sentence as the spread could indeed be attributed to the heterogenous nature of surface ocean P concentrations. In the modern ocean CAP varies by more than an order of magnitude in the surface ocean (Martiny Adam et al., 2019).

4. Has any petrography been performed on any of these samples? How were fabrics considered in sub-sampling for chemical analyses? Whether you are sampling a primary marine cement vs a late diagenetic cement vs an ooid that formed on the platform top can significantly change how CAP values should be interpreted, and by having data from the different formation and diagenetic environments, one can leverage these data to build a more complete picture of the environment across the LE.

3.14

Petrography has been performed on these samples in prior published studies (Kalderon-Asael et al., 2021; Ossa Ossa et al., 2018) before samples were selected for geochemical analysis. We subsampled the samples previously used in these other studies. When subsampling we avoided visible alteration or veining. In these prior studies detailed environmental reconstructions of the studied samples were performed. Despite widely varying depositional environments and a likely mix of primary and diagenetic cements the positive trend of d13C and CAP holds true. On lines 245-263, we discuss how this broad mix of environmental factors support a primary origin of the positive trend.

5. I appreciate how the authors investigated the CAP-d13C covariation from various angles in the SI. In my opinion, this paper would be stronger if it was published in a venue that allowed much of this detailed analysis and consideration to be moved into the main text, but that is just my opinion.

3.15

The SI file is intended for experts in the field who would find such information interesting but we believe the broad appeal of the findings beyond geochemists make this journal choice appropriate. We kept detailed diagenetic discussion in the SI so the general reader can easily digest the key takeaways.

Summary: I found this paper incredibly interesting and would like to see it published. My comments are intended to primarily serve to enrich the diagenetic assessment and boldness of some claims, but overall, I think the main finding is made on solid footing. Placing it within the context of a bit more diagenetic assessment would just ensure it stands the test of time.

3.16 We thank the reviewer for their support of this study and time spent reviewing and providing comments on this manuscript.

- Alcott, L. J., Mills, B. J. W., and Poulton, S. W., 2019, Stepwise Earth oxygenation is an inherent property of global biogeochemical cycling: *Science*, v. 366, no. 6471, p. 1333-1337.
- Alcott, L. J., Walton, C., Planavsky, N. J., Shorttle, O., and Mills, B. J. W., 2024, Crustal carbonate build-up as a driver for Earth's oxygenation: *Nature Geoscience*, v. 17, no. 5, p. 458-464.
- Canfield, D. E., Ngombi-Pemba, L., Hammarlund, E. U., Bengtson, S., Chaussidon, M., Gauthier-Lafaye, F., Meunier, A., Riboulleau, A., Rollion-Bard, C., Rouxel, O., Asael, D., Pierson-Wickmann, A.-C., and El Albani, A., 2013, Oxygen dynamics in the aftermath of the Great Oxidation of Earth's atmosphere: *Proceedings of the National Academy of Sciences*, v. 110, no. 42, p. 16736-16741.
- Dodd, M. S., Shi, W., Li, C., Zhang, Z., Cheng, M., Gu, H., Hardisty, D. S., Loyd, S. J., Wallace, M. W., vS. Hood, A., Lamothe, K., Mills, B. J. W., Poulton, S. W., and Lyons, T. W., 2023, Uncovering the Ediacaran phosphorus cycle: *Nature*, v. 618, no. 7967, p. 974-980.
- Dodd, M. S., Zhang, Z., Li, C., Algeo, T. J., Lyons, T. W., Hardisty, D. S., Loyd, S. J., Meyer, D. L., Gill, B. C., Shi, W., and Wang, W., 2021, Development of carbonate-associated phosphate (CAP) as a proxy for reconstructing ancient ocean phosphate levels: *Geochimica et Cosmochimica Acta*, v. 301, p. 48-69.
- Geyman, E. C., and Maloof, A. C., 2019, A diurnal carbon engine explains ¹³C-enriched carbonates without increasing the global production of oxygen: *Proceedings of the National Academy of Sciences*, v. 116, no. 49, p. 24433-24439.
- Halevy, I., and Bachan, A., 2017, The geologic history of seawater pH: *Science*, v. 355, no. 6329, p. 1069-1071.
- Ingalls, M., Blättler, C. L., Higgins, J. A., Magyar, J. S., Eiler, J. M., and Fischer, W. W., 2020, P/Ca in Carbonates as a Proxy for Alkalinity and Phosphate Levels: *Geophysical Research Letters*, v. 47, no. 21, p. e2020GL088804.
- Ingalls, M., Grotzinger, J. P., Present, T., Rasmussen, B., and Fischer, W. W., 2022, Carbonate-Associated Phosphate (CAP) Indicates Elevated Phosphate Availability in Neoproterozoic Shallow Marine Environments: *Geophysical Research Letters*, v. 49, no. 6, p. e2022GL098100.
- Kalderon-Asael, B., Katchinoff, J. A. R., Planavsky, N. J., Hood, A. v. S., Dellinger, M., Bellefroid, E. J., Jones, D. S., Hofmann, A., Ossa, F. O., Macdonald, F. A., Wang, C., Isson, T. T., Murphy, J. G., Higgins, J. A., West, A. J., Wallace, M. W., Asael, D., and Pogge von Strandmann, P. A. E., 2021, A lithium-isotope perspective on the evolution of carbon and silicon cycles: *Nature*, v. 595, no. 7867, p. 394-398.
- Martiny Adam, C., Lomas Michael, W., Fu, W., Boyd Philip, W., Chen Yuh-ling, L., Cutter Gregory, A., Ellwood Michael, J., Furuya, K., Hashihama, F., Kanda, J., Karl David, M., Kodama, T., Li Qian, P., Ma, J., Moutin, T., Woodward, E. M. S., and Moore, J. K., 2019, Biogeochemical controls of surface ocean phosphate: *Science Advances*, v. 5, no. 8, p. eaax0341.

- Ossa Ossa, F., Eickmann, B., Hofmann, A., Planavsky, N. J., Asael, D., Pambo, F., and Bekker, A., 2018, Two-step deoxygenation at the end of the Paleoproterozoic Lomagundi Event: *Earth and Planetary Science Letters*, v. 486, p. 70-83.
- Ren, C., Li, Y., Wang, S., Wang, J., Ji, J., Teng, H. H., Phillips, B. L., Kwon, K. D., and Li, W., 2025, Coprecipitation of phosphate with calcite: Molecular-scale evidence for incorporation and inclusion mechanisms: *Geochimica et Cosmochimica Acta*, v. 399, p. 1-17.
- Richardson, J. A., Roest-Ellis, S., Phillips, B. L., Strauss, J. V., Webb, S. M., and Tosca, N. J., 2022, Characterization and Geological Implications of Precambrian Calcite-Hosted Phosphate: *Geophysical Research Letters*, v. 49, no. 17, p. e2022GL100328.
- Roest-Ellis, S., Richardson, J. A., Phillips, B. L., Mehra, A., Webb, S. M., Cohen, P. A., Strauss, J. V., and Tosca, N. J., 2023, Tonian Carbonates Record Phosphate-Rich Shallow Seas: *Geochemistry, Geophysics, Geosystems*, v. 24, no. 5, p. e2023GC010974.

Reviewer #1 (Remarks to the Author):

I commend the authors on doing a good job responding to my earlier concerns, and I congratulate them on a thought-provoking paper. It will be interesting to see how work on the CAP proxy and the GOE evolves in the coming years.

Reviewer #2 (Remarks to the Author):

The authors took care to incorporate the comments from 3 reviewers into this revised version of the manuscript. With the altered wording, the authors to my opinion mostly addressed the main criticism of all three reviewers, namely the caution required when using a sedimentary proxy to reconstruct ancient seawater.

Of course, in paleoceanography large uncertainties inherently exist and all three reviewers agreed that Dodd et al. provide valuable new data that help shed further light on the co-evolution of the marine P cycle and atmospheric oxygenation.

In the revised discussion, Dodd et al. now include existing work regarding ancient marine P cycling (e.g. Reinhard et al., Canfield et al.) and paint a more nuanced picture of their results, which I appreciate. What I do note that is that the new level of caution in interpretation of results is not included in the title; I understand the wish for a snappy title, but I would recommend a less definitive statement, considering the amount of caution now expressed in the manuscript.

Regarding the link between CAP and the size of the oceanic dissolved P reservoir, I remain of the opinion that additional data such as TOC, total P, trace metals would contribute significantly to our understanding of the sedimentary and depositional context (and P cycling) that helps shape the CAP record as well. The authors consider this 'beyond the scope of this study' and it likely complicates the story, but I find it a missed opportunity to more fully develop our understanding of the P cycle at that time. Furthermore, the authors mention that "This study provides evidence for how ocean P level was changing, but not what was causing it to change" but my point is more that additional data would help better understand sedimentary P cycles that helps shape the CAP record. That being said, also considering the feedback from the other reviewers, I appreciate that Dodd et al. included more elaborate discussion on ancient P cycling from other studies.

Response: We added the following sentence on lines 344: Future work integrating CAP with Total Organic Carbon, total P and trace-metal datasets will help further disentangle P cycling controls on CAP and marine P variability.

For the modelling, appreciate that in the revised manuscript, Dodd et al. emphasize the evidence that suggest a "P-driven GOE" (C isotopic record, pO₂ reconstructions).

Overall, Dodd et al. provide a revised manuscript that includes a more nuanced and therefore stronger presentation and discussion of the data.

A minor point in the revised manuscript with tracked changes is that the phrasing in L789-791 is a bit confusing:

"Rising sulphate levels would have stimulated microbial sulphate reduction and enhanced P recycling from organic rich sediments. P speciation data from this interval support increased recycling efficiency under more oxidizing conditions due to sedimentary sulphur cycling"

This feels counterintuitive; P recycling is generally more efficient under more reducing sedimentary conditions, in this case more sulfide generation and subsequent trapping of Fe in sulphides. I think the authors refer to a more oxygenated atmosphere that would result in more sulfate supply to the oceans? It's probably useful to clarify these aspects a bit (both how enhanced sulphate reduction affects P burial efficiency and what part of the atmosphere-ocean water-seafloor system is more oxygenated).

Response: Modified text on lines 331 as follows: P speciation data from this interval support increased recycling efficiency as a result of increasing sulphate availability, due to sedimentary sulphur cycling ⁵⁰, reinforcing a positive feedback in which rising atmospheric oxygen levels increased sulphate availability in anoxic water bodies amplifying P release and sustained high productivity.

Reviewer #3 (Remarks to the Author):

I recommend publication of the revised version of this manuscript.

Review of Increased phosphorus bioavailability triggered Earth's permanent surface oxygenation by Dodd et al.

Overview: The authors present carbonate-associated phosphate (CAP) data from globally distributed marine sediments across the GOE and demonstrate a positive covariation with carbonate carbon isotopes. The primary finding of this analysis is that CAP values are high during the Lomagundi positive carbon isotope excursion. These data are used in a biogeochemical box model to support how increased P availability would lead to a rapid oxygenation of the atmosphere by increased primary productivity (photosynthesis).

Comments:

1. In several places, including a section title in the Introduction, the authors refer to CAP as a direct proxy for seawater phosphate concentrations. However, the first author's 2021 paper demonstrates a strong dependence of phosphate incorporation into carbonate minerals on pH and mineralogy. I recommend tempering the language around direct proxy if the evaluation is based on qualitative comparisons between CAP datasets across time without quantifying seawater [P]. I want to make clear that I think there is significant value in these qualitative comparisons.
2. I appreciate the approach of considering metamorphic grade in your diagenetic assessment of these samples. However, I question whether comparing across similar metamorphic grade is valid. Presumably, alteration of CAP values is more related to the phosphate contents of the diagenetic fluid rather than the P-T regime of the alteration environment (other than potentially extent of water-rock exchange). I recommend performing some amount of petrography on samples from each formation and analyzing a clear secondary fabric for CAP, such as a secondary vein. This will provide some information about the phosphate contents of altering fluids which will then provide a means of assessing whether CAP values in your fabrics interpreted as primary provide a minimum or maximum cap on CAP. A few analyses of secondary fabrics would really bolster your argument for accepting CAP values as primary, or at least usable for comparison across geographic regions.
3. Rationale for spread in CAP values: In the very first sentence of the Discussion, the authors state "There is a large spread in CAP values of formations deposited during the LE, which is expected given the heterogeneous nature of phosphorus concentration in the surface ocean." Can the spread realistically be explained by a reasonable range of surface seawater [P]? I think it is more likely pointing to variable degrees of water-rock alteration and/or phosphate contents of altering fluids in the metamorphic environment. To be clear, I don't think this takes away from your main finding that there is a positive covariance in CAP and $\delta^{13}\text{C}$ across the LE, but I recommend casting this within a more cautious light with respect to primacy of CAP values.
4. Has any petrography been performed on any of these samples? How were fabrics considered in sub-sampling for chemical analyses? Whether you are

sampling a primary marine cement vs a late diagenetic cement vs an ooid that formed on the platform top can significantly change how CAP values should be interpreted, and by having data from the different formation and diagenetic environments, one can leverage these data to build a more complete picture of the environment across the LE.

5. I appreciate how the authors investigated the CAP-d13C covariation from various angles in the SI. In my opinion, this paper would be stronger if it was published in a venue that allowed much of this detailed analysis and consideration to be moved into the main text, but that is just my opinion.

Summary: I found this paper incredibly interesting and would like to see it published. My comments are intended to primarily serve to enrich the diagenetic assessment and boldness of some claims, but overall, I think the main finding is made on solid footing. Placing it within the context of a bit more diagenetic assessment would just ensure it stands the test of time.